

# The influence of snow sublimation on stable isotopes of water vapor in the atmospheric boundary layer of Central Europe

Emanuel Christner[1], Martin Kohler[1], and Matthias Schneider[1]

[1]Institute of Meteorology and Climate Research, Karlsruhe Institute of Technology (KIT), Germany

*Correspondence to:* E. Christner (emanuel.christner@kit.edu)

**Abstract.** Post-depositional fractionation of stable water isotopes due to fractioning surface evaporation introduces uncertainty to various isotope applications such as the reconstruction of paleotemperatures, paleoaltimetry, and the investigation of ground-water formation. In this study, we investigate isotope fractionation during the evaporation of snow at ground level by combining 17 months of observations of isotope concentration ratios $[HD^{16}O]/[H_2^{16}O]$ in low-level water vapor in Central Europe with a

new Lagrangian isotope model. The isotope model is capable of reproducing variations of the observed isotope ratios with a correlation coefficient R of 0.82. Deviations between modeled and measured isotope ratios during cold snaps were related to differences in skin temperatures ($T_{\mathrm{skin}}$). Analysis of $T_{\mathrm{skin}}$ provided by the Global Data Assimilation System (GDAS) of the NCEP implies the existence of two regimes of $T_{\mathrm{skin}}$ with different types of isotope fractionation during evaporation: A cold regime with $T_{\mathrm{skin}} < T_{\mathrm{subl,max}} = -7.1°C$, which is dominated by non-fractioning sublimation of snow and a warmer regime with

$T_{\mathrm{subl,max}} < T_{\mathrm{skin}} < 0°C$, which is dominated by fractioning evaporation of meltwater. Based on a sensitivity study, we assess an uncertainty range of the determined $T_{\mathrm{subl,max}}$ of $-11.5$ to $-2.5°C$.

## 1 Introduction

The hydrological cycle of the atmosphere is usually investigated by water vapor concentration measurements. A new dimension is opened by the analysis of $H_2O$ stable isotope ratios, which are modified by $H_2O$ phase changes during evaporation,

cloud formation, and in-cloud physics. The main reason for this fractionation is the difference between vapor pressures of the stable isotopes $H_2^{16}O$ and $HD^{16}O$, which results in a preferential condensation of $HD^{16}O$. In consequence, not only specific humidity and dew point temperature but also the isotope concentration ratio $R_D = [HD^{16}O]/[H_2^{16}O]$ decreases in a cooling and raining air mass. The resulting relation between condensation temperature and isotope ratios of water vapor or precipitation is the basis for a variety of applications. Water isotopes in ice cores are used for the high-resolution reconstruction of pa-

leotemperatures (Dansgaard, 1964; Masson-Delmotte et al., 2005). The temperature-induced gradient of isotope ratios with altitude makes water isotopes in precipitation a proxy for paleotopography (Poage, 2001; Rowley et al., 2001; Blisniuk, 2005; Rowley and Garzione, 2007). Hydrological studies exploit the altitude-dependence and a seasonality of isotope ratios in precipitation to investigate groundwater formation (de Vries and Simmers, 2002).

Isotope fractionation during evaporation at the ground level post-depositionally modifies the isotope ratio of water at the sur-

face and in the upper soil layers (Barnes and Allison, 1983). Because the various isotope applications rely on a close relation


between the isotope ratio of precipitation and the isotope ratio of water at the surface, post-depositional effects increase the uncertainty of the isotope applications. Most problematic in that context are systematic post-depositional changes of isotope ratios into one direction, which most likely affect water intercepted in the snowpack because of long exposure to the atmosphere.

Several studies report post-depositional enrichment of heavy isotopes in the snowpack. Such enrichment was observed over a wide range of temperatures and even at skin temperatures far below the freezing point. Epstein et al. (1965) observed enhanced isotope ratios in depth-hoar layers in South Pole firn. To explain the observations, the authors suggested non-fractioning sublimation and subsequent fractioning recondensation of a part of the vapor combined with the escape of the other part of the vapor. Moser and Stichler (1974) analyzed surface layer snow from Switzerland (2450 m a.s.l.) during an eight-day fair weather period with air temperatures between about $-5$ and $0°C$. They observed a continuous increase of isotope ratios in the aging surface layer snow, which they attributed to fractioning evaporation or sublimation of snow. Stichler et al. (2001) report a similar experiment from the Chilean Andes (5536 m a.s.l.) with air temperatures between about $-12$ and $-5°C$. They observed an increase of isotope ratios in the aging surface layer snow as well. They attributed this increase to kinetic fractionation during sublimation, caused by smaller coefficients of diffusion of the heavier isotopes. Based on a seasonal increase of isotope ratios in a snowpack in northern Norway ($\sim$900 m a.s.l.), Gurney and Lawrence (2004) suggested fractioning evaporation of meltwater and subsequent recrystallization of residual meltwater. Consistent with this, Lechler and Niemi (2011) report an altitude-dependent seasonal modification of isotope ratios in an alpine snowpack, which they attributed to fractioning evaporation of meltwater during ablation.

A clear assignment of such observations to a specific type of interaction between snowpack and atmosphere or to certain meteorological conditions is difficult, as continuous time series of high-resolution isotope profiles of the snowpack are hard to obtain. Furthermore, processes such as meltwater percolation, diffusion, sublimation, and resublimation within the snowpack additionally modify the profiles of isotope ratios, making an interpretation even more challenging. The current understanding of fractionation during the evaporation of snow is therefore highly uncertain and stretches from non-fractioning layer-by-layer sublimation of snow without any modification of isotope ratios in the snowpack (Ambach et al., 1968; Dansgaard, 1973; Friedman et al., 1991) to systematic enrichment of heavy isotopes in the snow in consequence of fractioning evaporation of meltwater (Gurney and Lawrence, 2004; Lechler and Niemi, 2011).

In this context, continuous observations of isotope ratios of low-level water vapor may provide new insights by offering the opportunity to investigate fractionation during evaporation from the snowpack from a complementary point of view. A case study by Noone et al. (2013) investigates local variations of isotope ratios in water vapor subsequent to a winter storm. Based on observations on a research tower the study partitioned the different surface fluxes assuming local evapotranspiration to consist of non-fractioning sublimation and fractioning evaporation of meltwater.

A promising way to extend the investigation of isotope fractionation during evaporation of snow to the remote moisture source regions is the combination of isotope observations with Lagrangian isotope modeling. A number of earlier studies applied Lagrangian isotope modeling along idealized, climatological trajectories (Jouzel and Merlivat, 1984; Johnsen et al., 1989) or along individual back trajectories (Schlosser et al., 2004; Helsen et al., 2004, 2005, 2007; Sodemann et al., 2008a;





Pfahl and Wernli, 2009). As these studies mainly focused on polar or marine regions, the applied Lagrangian isotope models are not optimized to simulate continental evapotranspiration in the middle latitudes or evaporation of snow or meltwater at temperatures close to the freezing point.

In this paper, we present a time series with 17 months of continuous measurements of $R_D$ in low-level water vapor in Central Europe. The measurements cover two winters, which were marked by a number of cold snaps and related snowfall. By combining isotope observations during the cold snaps with a new Lagrangian isotope model, we investigate isotope fractionation during sublimation or evaporation of surface layer snow.

In the following, we present the Lagrangian isotope model (Sect. 2) and characterize the uncertainty of our isotope measurements (Sect. 3). In Sect. 4, we relate variations of the isotope ratios to air temperature and specific moisture source regions during cold snaps. In Sect. 5, isotope ratios during cold snaps are analyzed with respect to fractionation during evaporation or sublimation of snow at the surface.

## 2 Lagrangian moisture diagnostics

### 2.1 Back trajectories

Isotope ratios to be analyzed in this paper were measured at a site near Karlsruhe in Central Europe (49.10° N, 8.44° E, 110.4 m a.s.l.). Kinematic five-day back trajectories from the site were calculated with the Hybrid Single Particle Lagrangian Integrated Trajectory Model 4.0 (Draxler and Hess, 1998) with a time resolution of 1 h. Three-dimensional wind fields for trajectory calculation were derived from the Global Data Assimilation System (GDAS) of the National Centers for Environmental Prediction (NCEP) (Kanamitsu, 1989; Derber et al., 1991; Parrish and Derber, 1992). The GDAS data (V1.5) is available for every 3 h at a 1×1° horizontal resolution and on 23 sigma pressure levels between 1000 and 20 hPa. To account for uncertainty of the back trajectories, we used trajectory ensembles. Each ensemble consists of nine trajectories, starting 30 m above ground level at the measurement site as well as 50 km N, NE, E, SE, S, SW, W, and NW from the site. Back trajectories were calculated for every three hours of measurement time and were initialized at 0, 3, 6, 9, 12, 15, 18, and 21 UTC.

### 2.2 Lagrangian diagnostic of moisture sources

In order to identify major source regions of low-level water vapor in Karlsruhe, we applied a Lagrangian source region analysis similar to the method described by Sodemann et al. (2008b). The method traces air parcels along kinematic five-day back trajectories and analyzes changes of specific humidity ($q$) in time intervals of one hour. Such changes are possible due to the formation of precipitation ($P$), evaporation from the ground ($E$), evaporation of falling rain, air mass mixing due to convection or small-scale turbulence, diffusion, and numerical errors. Like Stohl and James (2004) and Sodemann et al. (2008b), we assume $P$ and $E$ to be the dominant processes and ignore the other effects. Using this simplification, the change of specific humidity ($\Delta q$) per time step ($\Delta t$) is:

$$\Delta q/\Delta t = (E - P)/\Delta t. \tag{1}$$





Assuming further that either $E$ or $P$ are dominating (James et al., 2004), the net change $\Delta q/\Delta t$ per time step is attributed to only $E$ or to $P$. Corresponding to these assumptions, an identified decrease of specific humidity is attributed to the formation of precipitation. In cases of a positive increment of specific humidity, the method assumes moisture uptake from evaporation at ground level. In this case, the contribution of surface evaporation ($f_m$) in a time interval $m$ to total specific humidity at the end of this time interval ($q_m$) is

$$f_m = \frac{\Delta q_m}{q_m}. \tag{2}$$

The formation of precipitation in a later time interval does not affect the $f_m$ calculated for the earlier time interval. In contrast to that, further moisture uptake in a later time interval $n$ reduces the relative contribution of surface evaporation in the earlier time interval $m$ to the moisture at the end of $n$. In this case, the contribution from earlier moisture uptake to $q_n$ is recalculated according to:

$$f'_m = f_m \cdot \frac{q_n - \Delta q_n}{q_n}. \tag{3}$$

Fast and random variations of $q$ along the back trajectories could result in an overestimation of the formation of precipitation and moisture uptake. To avoid such an overestimation, fast variations of $q$ were removed by smoothing $q$ along the back trajectories with a 24 h rectangle function. Arbitrarily choosing a width of 24 h may smooth out real diurnal variations of $q$ and thereby, may result in a displacement or underestimation of the amount of moisture uptake. Wherever necessary for the interpretation of our results, we therefore assess the related uncertainty of findings by changing the width of the rectangle function to 12 and 36 h.

In cases of moisture uptake above the atmospheric boundary layer an attribution to surface evaporation is not directly evident. A maximum altitude for consideration of moisture uptake might therefore be appropriate. However, Aemisegger et al. (2014) relate moisture uptake at higher levels for trajectories starting from Rietholzbach in northern Switzerland to the outflow of shallow convection. As moisture in Karlsruhe and Rietholzbach originates from similar source regions, we also do not assume a maximum altitude for the consideration of surface evaporation.

### 2.3 Lagrangian isotope model

A Lagrangian isotope model was developed to serve as a benchmark for our $\delta D$ measurements. Like the Lagrangian moisture source diagnostic, the model runs along kinematic five-day back trajectories and attributes changes of specific humidity ($q$) to the formation of precipitation or moisture uptake from surface evaporation. Analogous to the moisture source diagnostic, the model does not apply a maximum altitude for moisture uptake, smoothes $q$ along the trajectories with a 24 h rectangle function, and uses the same trajectory ensembles. From each trajectory ensemble, nine modeled $\delta D$ ratios for Karlsruhe are obtained, which are combined to one value by calculating the $q$-weighted average.



### 2.3.1 Dehydration

A decrease of specific humidity in a time interval indicates the formation of precipitation. During the formation of precipitation, condensation of $HD^{16}O$ is more likely than condensation of $H_2^{16}O$, because of $HD^{16}O$'s lower vapor pressure. The formation of precipitation therefore results in a decreasing isotope concentration ratio ($R_D$=[$HD^{16}O$]/[$H_2^{16}O$]) in an air mass. Assuming

equilibrium conditions during condensation and subsequent immediate rainout of the condensate, we calculate the change of isotope ratios of the residual water vapor according to the Rayleigh distillation model (Rayleigh and Ramsay, 1894). Based on a fractionation factor $\alpha_D(T)$, this Rayleigh model calculates changes of $R_D$ for infinitesimal changes of specific humidity $q$:

$$d\ln R_D = [\alpha_D(T) - 1] \cdot d\ln q \tag{4}$$

$\alpha_D$ depends on the temperature ($T$) of an air parcel and increases from about 1.082 to 1.209 between $+20°C$ and $-30°C$. For

$T{>}=0°C$ we use a parametrization of $\alpha_D$ over liquid water (Horita and Wesolowski, 1994). For $T{<}0°C$, we assume enhanced fractionation over ice and use the parametrization of Jancso and Van Hook (1974). Additional non-equilibrium ("kinetic") fractionation in the case of supersaturation in ice clouds (Merlivat and Jouzel, 1979; Jouzel and Merlivat, 1984) is ignored by the model because of moderate temperatures and correspondingly low degrees of supersaturation in the examined source regions.

### 2.3.2 Moistening

In the case that specific humidity increases in a time interval [$t_1$,$t_2$], we assume a permeable air parcel which takes up moisture $\Delta q$ by turbulent mixing. We attribute that moisture to evaporation at ground level with the isotope ratio $R_{D,E}$, which was lofted via small-scale turbulence or convection to the trajectory level. To calculate the corresponding change of $R_D$ of moisture in the tracked air parcel, we apply the following mixing equations:

$$q(t_2) = q(t_1) + \Delta q, \tag{5}$$

$$R_D(t_2) = \frac{R_D(t_1) \cdot q(t_1) + \Delta q \cdot R_{D,E}(t_1)}{q(t_1) + \Delta q}. \tag{6}$$

If the entrained moisture from evaporation was transported via small-scale turbulence to the trajectory altitude, dilution with air from below the trajectory level is likely. Applying the above equations, we therefore implicitly assume that the air masses below the trajectory level experienced a similar transport and precipitation history as the tracked air parcel. In this case, $R_D$

and $q$ of the tracked air parcel are not affected by dilution with that air from below.

Depending on the type of ground and skin temperature, we calculate $R_{D,E}$ assuming evaporation from the ocean ($R_{D,E\_ocean}$), continental evapotranspiration ($R_{D,E\_ET}$), evaporation of melted snow ($R_{D,E\_snowevap.}$), or sublimation of snow ($R_{D,E\_snowsubl.}$). For evaporation from the ocean, we assume equilibrium fractionation over liquid water according to the parameterization of Horita and Wesolowski (1994). Kinetic fractionation during evaporation on the order of $\alpha_{D,kin}$=1.002–1.007 (Merlivat and Jouzel,

1979; Pfahl and Wernli, 2009) due to different coefficients of diffusion of the different water isotopes is ignored by the model. Please note that $\alpha_{D,kin}$ is much smaller than the vapor pressure fractionation factor $\alpha_D$. Therefore, consideration of kinetic





fractionation is less important for $R_D$ than for other frequently used isotope ratios such as $[H_2^{18}O]/[H_2^{16}O]$. Given the above assumptions, the isotope ratio of evaporated moisture $R_{D,\text{E\_ocean}}$ only depends on the sea surface temperature (SST) and the isotope ratio of sea surface water $R_{D,\text{ocean}}$, which we derive from LeGrande and Schmidt (2006):

$$R_{D,\text{E\_ocean}} = \frac{R_{D,\text{ocean}}}{\alpha_D(\text{SST})}. \qquad (7)$$

Over the continent, evapotranspiration consists of evaporation from the bare soil, transpiration of plants, and evaporation from canopy interception. As a first simplification we ignore canopy interception, i.e. we consider condensation with subsequent complete re-evaporation as a neutral process with respect to $q$ and $R_D$. Moisture from the two other sources strongly differs in isotopic composition.

Evaporation from the bare soil is accompanied by isotope fractionation (Zhang et al., 2010). To calculate the isotope ratio of moisture evaporated from the bare soil ($R_{D,\text{E\_soilevap.}}$), we assume equilibrium fractionation. Kinetic fractionation during evaporation from the soil on the order of $\alpha_{D,\text{kin}}$=1.017–1.025 (Mathieu and Bariac, 1996) is ignored. For the calculation of $\alpha_D(T)$, we apply the parameterization of Horita and Wesolowski (1994) for equilibrium fractionation over liquid and use the skin temperature at the trajectory position. Since time of moisture uptake and evaporation might differ, we weighted skin temperatures along the back trajectories with surface evaporation within 24 h ($T_{\text{skin}}$). Because soil water around the measurement site is recharged by precipitation on average every 2.6 days, we ignore the enrichment of $HD^{16}O$ in the soil. We further assume the isotope ratio of soil water to be the same as the isotope ratio of precipitation ($R_{D,\text{prec.}}$), which we derive from the climatological monthly means of the Regional Cluster-based Water Isotope Prediction (RCWIP) (Terzer et al., 2013):

$$R_{D,\text{E\_soilevap.}} = \frac{R_{D,\text{prec.}}}{\alpha_D(T_{\text{skin}})}. \qquad (8)$$

In contrast to bare soil evaporation, plants take up soil water from the soil and transpire that water completely and therefore with no fractionation into the atmosphere. Still some fractionation is possible on short timescales, due to asynchronous accumulation and the release of $HD^{16}O$ and $H_2^{16}O$ in leaves (Zhang et al., 2010). However, this process averages out over a day (Harwood et al., 1999; Farquhar et al., 2007) and is therefore ignored for modeling isotope ratios along the five-day back trajectories. To calculate the isotope ratio of moisture originating from plant transpiration, we assume:

$$R_{D,\text{E\_transp.}} = R_{D,\text{prec.}}. \qquad (9)$$

The isotope ratio of total evapotranspiration ($R_{D,\text{E\_ET}}$) depends on the fraction of plant transpiration (FT) on total evapotranspiration:

$$\begin{aligned} R_{D,\text{E\_ET}} &= R_{D,\text{E\_soilevap.}} \cdot (1 - \text{FT}) + R_{D,\text{E\_transp.}} \cdot \text{FT} \\ &= \frac{R_{D,\text{prec.}}}{\alpha_D(T_{\text{skin}})} \cdot (1 - \text{FT}) + R_{D,\text{prec.}} \cdot \text{FT}. \end{aligned} \qquad (10)$$

FT varies with region and on seasonal, synoptic, and diurnal timescales. For modeling we ignore these variations and use a constant fraction of transpiration. Based on Choudhury et al. (1998), Lawrence et al. (2007), and Aemisegger et al. (2014) we





assume an average FT in Europe of 0.7.

Whenever we observe moisture uptake at continental skin temperatures below $0°C$, we ignore transpiration of plants (FT=0) and attribute the moisture to the evaporation of melted snow or ice. In this case, we again assume equilibrium fractionation over liquid. We further assume the isotope ratio of snow to be the same as the climatological monthly means of the RCWIP:

$$R_{D,\text{E\_snowevap.}} = \frac{R_{D,\text{prec.}}}{\alpha_D(T_\text{skin})}. \tag{11}$$

In Sect. 5 we investigate a possible role of snow sublimation. In this case we define a skin temperature $T_\text{subl,max}$. Below that temperature we assume complete layer-by-layer sublimation of snow without isotope fractionation:

$$R_{D,\text{E\_snowsubl.}} = R_{D,\text{prec.}}. \tag{12}$$

### 2.3.3 Initialization

Forty-three percent of the analyzed air masses originated from oceanic regions and altitudes below $2\,\text{km}$. Forty-two percent originated from the continent and also from altitudes below $2\,\text{km}$ above ground level. For the initialization of $R_D$ of these air masses, we assume isotope ratios ($R_{D,\text{ini}}$) in a convectively well-mixed atmospheric boundary layer (ABL), where water vapor and ocean surface water or soil water are in isotopic equilibrium:

$$R_{D,\text{ini},<2\text{km}} = \frac{R_{D,\text{ocean|prec.}}}{\alpha_D(\text{SST}|T_\text{skin})}. \tag{13}$$

For initialization at surface temperatures above $0°C$, we calculate fractionation factors according to the parametrization of Horita and Wesolowski (1994). At skin temperatures below $0°C$ we assume fractionation over ice and apply the parametrization of Jancso and Van Hook (1974).

Fifteen percent of air masses originated from an altitude ($h$) higher than $2\,\text{km}$ above ground level. For these air masses we assume a linear decrease of $R_{D,\text{ini}}$ from boundary layer ratios at an altitude of $2\,\text{km}$ to $R_{D,10\,\text{km}}$ of 0.45 (Hanisco et al., 2007; Sayres et al., 2010) at an altitude of $10\,\text{km}$.

$$R_{D,\text{ini},>2\text{km}} =$$
$$R_{D,\text{ini},<2\text{km}} - (R_{D,\text{ini},<2\text{km}} - R_{D,10\,\text{km}})\frac{h - 2\,\text{km}}{10\,\text{km} - 2\,\text{km}}. \tag{14}$$

The dependence of modeled isotope ratios from initialization decreases with moisture uptake along the back trajectories (Fig. 1). The uncertainty of $R_{D,\text{ini}}$ is especially strong at high altitudes, where air masses are strongly dehydrated and have a long history of isotope fractionation. Because air masses from high altitudes take up a lot of humidity during descent and transport to Karlsruhe, the uncertainty of $R_D$ from initialization is strongly reduced in Karlsruhe. Back trajectories corresponding to smaller moisture uptake typically originate from the ABL. $R_D$ in Karlsruhe of these back trajectories depends more strongly on $R_{D,\text{ini}}$, which on the other hand is much better defined within the ABL. Whenever it is necessary for the interpretation of our results, we assess related uncertainty of the modeled $\delta D$ by changing $R_{D,\text{ini}}$ in different model runs.





## 3 Measurements

### 3.1 Isotope water vapor measurements

The concentrations of $H_2^{16}O$ and $HD^{16}O$ in low-level water vapor were measured for 17 months on a research campus 12 km north of Karlsruhe in Southwest Germany (49.10° N, 8.44° E, 110.4 m a.s.l.).

For the continuous measurements we used a Picarro water isotopologue analyzer L2120-i, which analyzed the ambient water vapor with a sampling rate of 0.6 Hz. The measurement technique is based on cavity ring-down spectroscopy, where the beam of a tunable diode laser is directed through a cavity, filled with the air to be analyzed. Based on the ring-down time of the laser light intensity, absorption spectra are measured between 7183.5 and 7184 cm$^{-1}$. A characterization of two similar analyzers (L1115-i and L2130-i) can be found in Aemisegger et al. (2012). Please note that the isotopologue analyzer also measures
concentrations of the water isotope $H_2^{18}O$. As the isotope concentration ratio $R_{^{18}O}$=[$H_2^{18}O$]/[$H_2^{16}O$] is more sensitive to kinetic fractionation than $R_D$, it would be essential to consider kinetic fractionation during evaporation for modeling $R_{^{18}O}$ with the Lagrangian isotope model. Uncertainty of the modeled $R_{^{18}O}$ related to uncertainty of the kinetic fractionation factor does not allow deeper insight from analysis of $R_D$ and $R_{^{18}O}$ than from analysis of $R_D$ alone. Thus, we do not use the $H_2^{18}O$ measurements in this paper.

The Picarro water isotopologue analyzer was located on the sixth floor of the Institute of Meteorology and Climate Research – Atmospheric Trace Gases and Remote Sensing of the Karlsruhe Institute of Technology. A downward-facing inlet funnel was installed one meter above the edge of the roof, which corresponds to an altitude above ground level of 28 m. The connection to the inlet was established with a 6 m long tube with a diameter of 6.4 mm. To reduce wall effects, we permanently flushed the inlet line with 30 slpm of air and used tubing made of electropolished stainless steel. To avoid condensation, the wall temperature
of the 5 m of tubing inside the building was regulated to 22°C. Saturation humidity corresponding to this temperature is above the analyzer limit of 14.9 g·kg$^{-1}$. On five days in August 2012, humidity slightly exceeded that analyzer limit. Corresponding measurements were removed from the time series.

We report isotope measurements in a $\delta$-notation, which normalizes isotope ratios to a standard scale, defined by the Vienna standard mean ocean water (VSMOW: $\delta D$=0‰) and standard light Antarctic precipitation (SLAP: $\delta D$=−428.0‰) (IAEA,
2009):

$$\delta D = \frac{R_D}{R_{D,\text{standard}}} - 1. \tag{15}$$

For the automated calibration of the analyzer we applied a Picarro standard delivery module (A0101), which allows the alternating injection of two different water standards into a Picarro vaporizer (A0211). In this vaporizer the liquid standards immediately evaporate in a constant flow of dry synthetic air (140°C, 0.3 slpm dry air flow with 1.2 mg·kg$^{-1}$ residual humid-
30 ity). A two-point calibration was done for 2 h every 10 h at $\delta D$ of −62.1‰ and −142.2‰.

Instrumental drift during the 17 months was below 3‰ (Fig. 2a). The total accuracy of our $\delta D$ measurements due to uncertainty of calibrations and instrumental drift between two calibrations was 0.98‰. The 0.6 Hz precision of the measurements was below 1‰ and can be ignored for 10-minute averages shown in this paper.



Based on the two-point calibrations, we applied linear stretching to the measurements. Sixty-three percent of our observations are within the range of isotope ratios covered by the two standards. To approve linearity of the applied correction for isotope ratios below that range, we performed repeated calibrations with a third standard ($\delta D$=−245.3‰) in the two years subsequent to the campaign. We found additional uncertainty of $\delta D$ at −245.3‰ due to a slight non-linearity of the applied correction to

5 be smaller than 0.3 ‰, which is in agreement with the more detailed characterization of Aemisegger et al. (2012).

To identify a potential humidity dependence of the isotope ratio measurements, we generated three humidity levels between $1.8 \, \text{g} \cdot \text{kg}^{-1}$ and $13.7 \, \text{g} \cdot \text{kg}^{-1}$ during each calibration. We found the humidity dependence of the instrument to be smaller than the uncertainty of individual calibrations. Therefore, we only applied the average humidity dependence of the whole calibrations of $-0.021 \, \text{‰}/(\text{g} \cdot \text{kg}^{-1})$ to the data set (Fig. 2b).

**3.2 Meteorological data at the measurement site**

Observations of specific humidity at the measurement site were derived from the the Picarro isotope analyzer. For calibration of the Picarro humidity measurements we used observations of a VPT6 Thygan dew point mirror hygrometer (Meteolabor, Switzerland), which was mounted on a meteorological tower 30 m above ground level 900 m WSW. Since the topography at the measurement site is flat for some kilometers in all directions, we assume the tower observations to be representative

for the measurement site. The dew point hygrometer performed a measurement of one minute every ten minutes and has an uncertainty of $\pm 0.1 \, \text{K}$. Ten-minute averages of specific humidity derived from the Picarro Analyzer and observations of the dew point hygrometer show a correlation coefficient R of 0.9913. For the calibration of the Picarro humidity observations we applied the mean linear regression between hygrometer data and Picarro measurements of 1.13, which varied by 1% between the first and second half of the measurement period.

The amount of precipitation was measured at the meteorological tower at ground level with a time resolution of ten minutes.

**3.3 Isotope ratios in the moisture source regions**

Observations of $\delta D$ and $\delta^{18}O$ of ocean surface water are collected in the the Global Seawater Oxygen-18 Database (Schmidt et al., 1999). Since little data with $\delta D$ of ocean surface water exists, we calculated a median $\delta D/\delta^{18}O$ ratio of 6.56 from Fröhlich et al. (1988), Duplessy (1970), Delaygue et al. (2001), Gat et al. (1996), Ostlund et al. (1987), Aharon and Chappell (1986), Yobbi

(1992), and Weiss et al. (1979) and used this ratio to calculate $\delta D$ from $\delta^{18}O$ data. The $\delta^{18}O$ of ocean surface water along the back trajectories was derived from the spatial $1\times1°$ interpolation (V1.1) of the Global Seawater Oxygen-18 Database by LeGrande and Schmidt (2006) (Fig. 3a), which we linearly interpolated to the locations along the trajectories.

For the estimation of $\delta D$ of soil water along the back trajectories, we assumed soil water to have the same isotopic composition as precipitation. Observations of isotope ratios in precipitation have been collected in the Global Network of Isotopes in

Precipitation (GNIP) (Araguas et al., 1996) of the IAEA since the 1960s. We use climatological monthly means of $\delta D$ of the Regional Cluster-based Water Isotope Prediction (RCWIP) (Terzer et al., 2013), which provides a spatial interpolation of the GNIP data. RCWIP data is available with a horizontal resolution of $0.17\times0.17°$ (Fig. 3b/c) and was linearly interpolated to the locations along the trajectories.



### 3.4 Meteorological data in the moisture source regions

Specific humidity and air temperatures along back trajectories were derived from the identical GDAS data set used for the calculation of the back trajectories.

Skin temperatures ($T_{\mathrm{skin,unweighted}}$), accumulated surface latent heat flux, and a flag for snow cover along the back trajectories were derived from a reduced GDAS data set with the same horizontal resolution of $1 \times 1°$ but with data only every $6\,\mathrm{h}$. The data was interpolated linearly in space and time to the locations along the trajectories. The accumulated surface latent heat flux from the GDAS was divided by six to account for the hourly resolution of the trajectories. To derive skin temperatures representative of conditions during maximum evaporation ($T_{\mathrm{skin}}$), we weighted the $T_{\mathrm{skin,unweighted}}$ along the trajectories with positive surface latent heat flux in time intervals of $\pm 12\,\mathrm{h}$ (Fig. 4). If there were less than 12 trajectory points (time resolution of 1 h) with significant latent heat flux above $2\,\mathrm{W\,m^{-2}}$ in an interval, it was extended for $\pm 1\,\mathrm{h}$ until it contained 12 data points.

## 4 Analysis of seasonal and synoptic variations

In this section, we identify specific circulation regimes related to cold snaps in Karlsruhe. Subsequent to this, we examine the capability of the Lagrangian isotope model of reproducing corresponding variations of $\delta D$.

### 4.1 Source regions of moisture

For the identification of major source regions of low-level water vapor in Karlsruhe we applied the Lagrangian moisture source diagnostic described in Sect. 2.2. During all seasons the North Atlantic was the most important moisture source (Fig. 5), from where westerlies transported air masses to the measurement site. Due to a weakening of zonal circulation in summer, the source region was smallest in this season and concentrated on the eastern part of the North Atlantic. In winter, the source region was more extended, in consequence of a more pronounced zonal circulation, and expanded as far as the East Coast of the United States. In addition, occasional inversions of zonal circulation in winter led to easterly moisture transport.

### 4.2 Continental temperatures and zonal circulation

On a seasonal timescale air temperatures, specific humidity, and $\delta D$ in Karlsruhe followed a similar pattern (Fig. 6). In winter (DJF), air temperatures $30\,\mathrm{m}$ above ground level ($T$) were on average $2°\mathrm{C}$. Towards summer (JJA), $T$ increased to on average $20°\mathrm{C}$. Higher $T$ in summer corresponds to a higher saturation vapor pressure. That allows the transport of marine air to Karlsruhe with less condensation in summer than in winter. Consequently specific humidity ($q$) in Karlsruhe rose from $6\,\mathrm{g \cdot kg^{-1}}$ (DJF) to $14.8\,\mathrm{g \cdot kg^{-1}}$ (JJA). $\delta D$ changed from $-162‰$ (DJF) to $-109‰$ (JJA) and thereby consistently with $T$ and $q$ implies a lower degree of condensation and rainout in summer than in winter.

The gray color in Fig. 6 identifies circulation regimes with easterly moisture transport. Such regimes predominantly occurred in winter and resulted in the transport of continental air masses to Karlsruhe. The corresponding air masses usually were marked by especially low $T$ and $q$, which led to pronounced cold snaps in Karlsruhe. Consistent with the low $T$ and $q$, the air masses





during cold snaps showed an especially low $\delta D$ ratio.

Both findings – the seasonality of $\delta D$ and the especially low $\delta D$ in cold, continental air masses from the East – are in good agreement with the well-known "continental effect". That effect describes a decrease of $\delta D$ in precipitation over continents with distance to the coast (Fig. 3b/c), caused by the relation between $\delta D$ and condensation temperature. Since the $\delta D$ of rain depends on the $\delta D$ of the water vapor it is formed from, it is reasonable to find a similar continental effect imprinted to water vapor as well.

### 4.3   Comparison of measured and modeled $\delta D$

The Lagrangian isotope model described in Sect. 2.3 is able to reproduce the observed slow seasonal variation of $\delta D$, as well as the strong and relatively fast variations of $\delta D$ due to circulation regimes with predominantly easterly moisture transport in winter (Fig. 7a/b). The mean difference between modeled and observed $\delta D$ ($\Delta\delta D$) is +1.0‰. The correlation coefficient R of modeled and observed $\delta D$ is 0.82. Thereby, the correlation calculated for different seasons strongly differs from the overall correlation. For summer R is only 0.06 due to the small variability of $\delta D$ in this season. For winter R is 0.87.

Figure 7c shows the scatter plot of measured and modeled $\delta D$. Furthermore, the figure illustrates the impact of the formation of precipitation and surface evaporation on the modeled $\delta D$. If the formation of precipitation is ignored in the model, the overall correlation with the observations is still 0.81. The main reason for this is the relation of observations with low $\delta D$ ratios to easterly moisture transport. Corresponding back trajectories are initialized with relatively low $\delta D$ ratios according to GNIP observations in the respective continental source regions. This means that we only ignore the formation of precipitation in the five days covered by the back trajectories, but we implicitly consider the formation of precipitation which determined the isotope ratios in precipitation in the moisture source regions. However, due to surface evaporation with relatively high $\delta D$ ratios, the $\Delta\delta D$ in this scenario is +29.4‰. If one considers the formation of precipitation but ignores the surface evaporation the overall R is reduced to 0.62. The corresponding $\Delta\delta D$ is −35.6‰. So consideration of both processes, the formation of precipitation and surface evaporation is essential for reproducing the observed $\delta D$.

An interesting feature in Fig. 7a are spikes of low $\delta D$ ratios (blue), which are not reproduced by the model (red). We frequently observed such spikes from spring to autumn. Potential processes causing the spikes are the evaporation of rain below the cloud base or isotope exchange between falling raindrops and the low-level water vapor. An impact of these sub-cloud processes on the $\delta D$ of low-level water vapor is demonstrated for individual weather fronts by Wen et al. (2008) and Aemisegger et al. (2015) and is complementarily supported by observations of isotope ratios in precipitation (Friedman et al., 1962; Stewart, 1975; Gedzelman and Arnold, 1994). As isotope processes below clouds are not represented in the Lagrangian isotope model, the observations related to sub-cloud processes can't be further investigated by means of this model. However, a relevant role of sub-cloud processes for our observations of $\delta D$ spikes is supported by the strongly increased probability of precipitation during the spikes. For respective observations with $\delta D$ ratios smaller than expected from the standard deviation between observed and modeled $\delta D$ (23.5‰) the probability to observe precipitation in Karlsruhe within ±3 h was 44%, whereas for the other observations this probability was only 24%.





For the investigation of $\delta D$ during cold snaps in winter sub-cloud processes can be ignored, because the interaction between falling precipitation and water vapor is strongly suppressed in cases of solid precipitation.

## 5 $\delta D$ during cold snaps

The good agreement of modeled and measured $\delta D$ in winter (R=0.87) underlines the strong potential of the Lagrangian isotope model for analyzing isotope processes in the remote moisture source regions during cold snaps. In this section, we select respective observations and investigate isotope fractionation during sublimation or evaporation at ground level.

### 5.1 Sublimation of snow or evaporation?

Evaporation below $0°C$ was historically often considered as non-fractioning layer-by-layer sublimation of snow and ice (Ambach et al., 1968; Dansgaard, 1973; Friedman et al., 1991), because of a low coefficient of self-diffusion of water molecules in ice. However, this assumption ignores snow melt and fractionation during evaporation from the liquid phase, as implied by Gurney and Lawrence (2004), Lechler and Niemi (2011), and Noone et al. (2013). Both assumptions result in very different $\delta D$ of moisture from evaporation. In the case of non-fractioning sublimation, this $\delta D$ equals the $\delta D$ of the snow. In the case of fractioning evaporation, the $\delta D$ ratio of moisture from evaporation is about 90‰ lower. Our trajectory model provides an opportunity to test both formulations and to assess which one is more reasonable for Central Europe.

For the investigation of isotope fractionation during evaporation at low skin temperatures, we used data fulfilling two selection criteria. First, we excluded observations which are strongly affected by evapotranspiration at skin temperatures ($T_{\text{skin}}$) above the freezing point. For this purpose, we identified moisture uptake at $T_{\text{skin}}>0°C$ by means of the Lagrangian moisture source diagnostic and excluded air masses with a respective contribution above 2%. Second, we ensured relevant moisture uptake at $T_{\text{skin}}<0°C$. To this end, we identified moisture uptake at $T_{\text{skin}}<0°C$. For air masses meeting the first criterion the median contribution of moisture evaporated at $T_{\text{skin}}<0°C$ is 28%. For a meaningful interpretation, we only used the half with a contribution higher than 28%.

Some 178 of the three-hourly modeled data points meet both selection criteria. They belong to 38 different days and were used for further interpretation. Respective air masses mainly originated from the East (Fig. 8) and from altitudes below 2000 m above ground level. The GDAS data indicates the existence of snow on the ground at 96% of locations along the selected back trajectories. The median $T_{\text{skin}}$ at the trajectory positions five days back was $-11.7\pm5.2°C$ ($\pm$ gives the interquartile range). During transport to Karlsruhe the $T_{\text{skin}}$ rose on average by $6.0\pm3.4\,\text{K}$. A decrease of relative humidity due to warming of the air masses was partially compensated by moisture uptake and a corresponding increase of specific humidity by on average $52\pm35\%$.

Assuming equilibrium fractionation during evaporation of meltwater at $T_{\text{skin}}<0°C$ in the reference run ($M_{\text{MW}}$, Table 1), the model underestimates the selected $\delta D$ ratios by on average $\Delta\delta D=-19.7\pm1.6‰$ ($\pm$ gives the statistical uncertainty of the mean). Assuming non-fractioning sublimation at $T_{\text{skin}}<0°C$ in a further model run ($M_{\text{S}}$, Table 1) results in $\delta D$ ratios that are on average $\Delta\delta D=+25.7\pm1.7‰$ above the observations.





Considering the relatively high $\delta D$ ratios of moisture from non-fractioning sublimation and the about 90‰ lower $\delta D$ ratios in the case of fractioning evaporation, the difference of mean $\delta D$ between both model runs is qualitatively reasonable. To judge if one run provides more realistic results, we tested if one of the two scenarios can be brought into agreement with the observations when considering uncertainty of side constraints in the model.

5 As the most important side constraints for modeling the $\delta D$ in Karlsruhe, we consider interannual variability of $\delta D$ in the moisture source regions. Such variability is not captured by the model and therefore may systematically affect (1) the assumed $\delta D$ of snow as well as (2) $\delta D$ at the initialization. In addition, (3) the amount of identified moisture uptake needs to be accurate to reliably simulate the impact of sublimation or evaporation of snow on the $\delta D$ of water vapor.

In the following, we estimate uncertainty of these side constraints. Subsequent to this, we vary the side constraints in different 10 model runs, to assess corresponding systematic uncertainty of the modeled $\delta D$.

(1) The $\delta D$ of moisture from sublimation or evaporation of snow depends on the $\delta D$ of the snowpack, which we assume to be equal to the $\delta D$ of precipitation. The $\delta D$ of precipitation in an individual year may systematically differ from the used climatological monthly means of the Regional Cluster-based Water Isotope Prediction (RCWIP). To assess typical interannual variability of $\delta D$ of precipitation in winter, we used data from 134 European and Russian GNIP stations from the mid-latitudes 15 between 8.4 and 50° E with observations from at least three years. For each of the stations we calculated the mean $\delta D$ in the different winters (November, December, January, and February) and the standard deviation of the winter averages. The mean of standard deviations of the different stations was 11.5‰, which we assume to reflect the mean interannual variability of $\delta D$ in precipitation in the moisture source regions related to cold snaps.

In addition, the average monthly $\delta D$ of snow may differ from the average $\delta D$ of the monthly total precipitation recorded 20 by the GNIP. Because there is a general relation between surface air temperatures and $\delta D$ of precipitation in Central Europe (Schoch-Fischer et al., 1983; Jacob and Sonntag, 1991), winter months from years with especially high air temperatures and a potentially strong contribution from liquid precipitation are related to relatively high $\delta D$ ratios. Since we want to estimate $\delta D$ of the snowpack, data from winters with especially low temperatures and a strong contribution from solid precipitation with low $\delta D$ ratios is more suitable. During these winters the $\delta D$ of precipitation is probably closest to $\delta D$ of the RCWIP minus 25 the 11.5‰.

To test whether the too high modeled $\delta D$ ratios from the scenario of sublimation ($M_S$) can be significantly reduced when considering the systematic uncertainty of $\delta D$ of the snowpack, we performed one model run ($M_{S-,snow}$), in which we shifted $\delta D$ of snow by $-11.5$‰. To test whether the too low modeled $\delta D$ ratios from the scenario of evaporation of meltwater ($M_{MW}$) can be increased, we performed one further model run, in which we shifted $\delta D$ of snow by $+11.5$‰, although a positive shift 30 is not as likely as a negative shift. The mean difference between modeled and observed $\delta D$ ($\Delta\delta D$) of the different model runs is listed in Table 1.

(2) During cold snaps in Karlsruhe, on average 48% of humidity could be attributed to moisture uptake along the five-day back trajectories. The other side of this argument is that the $\delta D$ of 52% of humidity in Karlsruhe is determined by the initialization of isotope ratios.

35 For initialization below 2 km above ground level, we assume $\delta D$ in a well-mixed atmospheric boundary layer and isotopic





equilibrium between water vapor and the climatological monthly $\delta D$ of precipitation. This assumption is in agreement with one of the rare extended, simultaneous time series of $\delta D$ in water vapor and precipitation, conducted 45 km NNE from our site (Jacob and Sonntag, 1991). This study shows monthly averages of $\delta D$ in precipitation and water vapor at ground level for the years 1981 to 1988. The average deviation of the $\delta D$ of water vapor to isotopic equilibrium with precipitation in November,

December, January, and February was $+4.1\pm7.7‰$ ($\pm$ states the standard deviation calculated from the winter averages of the different years). Ignoring the interannually varying deviation between $4.1-7.7=-3.6‰$ and $4.1+7.7=+11.8‰$ from the isotopic equilibrium may systematically bias $\delta D$ at the model initialization.

For air masses originating from altitudes higher than 2 km above ground level, the uncertainty of $\delta D$ at initialization might be higher. Our model assumes a linear decrease of $\delta D$ ratios to $-550‰$ at an altitude of 10 km. Since in situ observations of $\delta D$

from the free troposphere are rare, uncertainty of this assumption is hard to assess. We assume that a systematic deviation of $\delta D$ to the estimated profile is smaller than $\pm100‰$. During cold snaps only 2% of the back trajectories originate from altitudes higher than 2 km above ground level, so that uncertainty of the average modeled $\delta D$ in Karlsruhe is only slightly affected by the uncertainty of an initialization in high altitudes.

To test how much the values of $\Delta\delta D$ for $M_\mathrm{S}$ and $M_\mathrm{MW}$ can be reduced if considering the uncertainty of $\delta D$ at the model

initialization, we performed two further model runs. For the scenario of sublimation we performed a model run in which we shifted $\delta D$ at the initialization for $-3.6‰$ / $-100‰$ in cases of initialization below / above 2 km above ground level ($M_\mathrm{S-,ini}$). For the scenario of evaporation of meltwater we shifted $\delta D$ at the initialization for $+11.8‰$ / $+100‰$ in cases of initialization below / above 2 km above ground level ($M_\mathrm{MW+,ini}$).

(3) To avoid misinterpretation of fast and random variations of specific humidity along the back trajectories as the formation

of precipitation and moisture uptake, we smoothed humidity with a 24 h broad rectangle kernel. Arbitrarily assuming a width of 24 h may result in an overestimation or underestimation of the formation of precipitation and moisture uptake. To assess the potential impact of the smoothing on the modeled $\delta D$, we changed the width of the applied rectangle kernel to 12 h in $M_\mathrm{S-,upt.12h}$ and to 36 h in $M_\mathrm{MW+,upt.36h}$.

To finally assess the minimum possible values of $\Delta\delta D$ in the case of superposition of the three sources of uncertainty discussed

above, we combined the assumptions of $M_\mathrm{S-,snow}$, $M_\mathrm{S-,ini}$, and $M_\mathrm{S-,upt.12h}$ in the model run $M_\mathrm{S---}$ and $M_\mathrm{MW+,snow}$, $M_\mathrm{MW+,ini}$, and $M_\mathrm{MW+,upt.36h}$ in the model run $M_\mathrm{MW+++}$.

Table 1 summarizes $\Delta\delta D$ for the different model runs. None of the model runs considering only one source of uncertainty is able to reduce $\Delta\delta D$ for the scenarios of sublimation or evaporation of meltwater to values close to 0. Even for $M_\mathrm{S---}$, which simultaneously assumes all the uncertainties of side constraints in the scenario of sublimation, $\Delta\delta D$ is $+13.5‰$. The discussed

uncertainty terms are therefore not able to bring model and observations into agreement with each other, if only considering non-fractioning sublimation. This implies that fractioning evaporation of meltwater played a significant role during our observations.

For $M_\mathrm{MW+++}$, which simultaneously assumes all the uncertainties of side constraints in the scenario of evaporation of meltwater, the value of $\Delta\delta D$ is reduced to 1.3‰. Considering the statistical uncertainty of $\Delta\delta D$ of 1.6‰, the average modeled

and measured $\delta D$ of the 178 selected air masses may therefore be brought into rough agreement with each other, if assuming





fractioning evaporation of meltwater. However, this requires superposition of the different uncertainty terms.

In order to refine this result, we split the selected observations into two groups of equal size according to the predominant skin temperature during moisture uptake ($T_{\text{skin,predom.}}$). For this purpose, we weighted skin temperatures along the individual ensembles of back trajectories with moisture uptake identified by the Lagrangian moisture source diagnostic. The median

$T_{\text{skin,predom.}}$ of the 178 selected trajectory ensembles is $-6.92°$C. We attributed data points to a group "cold", if $T_{\text{skin,predom.}}$ of the respective trajectory ensemble is below $-6.92°$C. For $T_{\text{skin,predom.}}$ above $-6.92°$C we attributed data points to a group "warm". Please note that also points of group "warm" have a $T_{\text{skin,predom.}}$ below $0°$C according to our selection criteria. Due to interannual variability of $T_{\text{skin,predom.}}$, data of the two groups is not randomly distributed in time. Seventy-six percent of group "cold" correspond to an especially pronounced cold snap in February/March 2012, whereas 85% of group "warm" belong to

cold snaps between October 2012 and February 2013.

Figure 9 shows two-dimensional probability distributions of the selected modeled and measured $\delta D$. Blue denotes data from group "cold" and red denotes data from group "warm". Under the assumption of non-fractioning sublimation ($M_S$), modeled $\delta D$ ratios of both groups are significantly higher than the observed values (Fig. 9a). The overestimation of modeled $\delta D$ ratios is especially strong in the regime with higher $T_{\text{skin}}$, where snow melt and fractioning evaporation would be expected. Figure

9b shows the respective probability distributions under the assumption of snow melt and fractioning evaporation of meltwater ($M_{\text{MW}}$). Under this assumption, the modeled $\delta D$ of group "warm" is close to the observations. However, modeled $\delta D$ ratios of group "cold" are now far too low.

Table 1 lists the mean differences between modeled and measured $\delta D$ for the two groups ($\Delta\delta D_{\text{cold}}$, $\Delta\delta D_{\text{warm}}$). As $\delta D$ of the individual groups may deviate more from the observations than the mean $\delta D$ of all 178 selected air masses, analyzing $\Delta\delta D_{\text{cold}}$

and $\Delta\delta D_{\text{warm}}$ allows drawing less ambiguous conclusions than analysis of $\Delta\delta D$. For $M_{S,---}$ the value of $\Delta\delta D_{\text{warm}}$ (16.7‰) is larger than the value of $\Delta\delta D$ of all selected data (13.5‰) which underlines the importance of fractioning evaporation for reproducing the observations. For $M_{\text{MW},+++}$ the value of $\Delta\delta D_{\text{cold}}$ (11.9‰) is larger than the value of $\Delta\delta D$ (1.3‰). So even $M_{\text{MW},+++}$, in which we simultaneously assumed all the uncertainties of side constraints in the scenario of fractioning evaporation of meltwater doesn't allow reproducing the observations of group "cold". This, in turn, implies significant non-

fractioning sublimation during our observations.

Comparison of $\delta D$ observations with $\delta D$ of the Lagrangian isotope model therefore implies a relevant role of both types of isotope fractionation in Central Europe.

## 5.2    Temperature-dependent types of fractionation

To simultaneously bring into agreement modeled and observed $\delta D$ of both groups, we suggest the existence of two regimes of

$T_{\text{skin}}$ with predominant non-fractioning sublimation in the colder regime and predominant fractioning evaporation of meltwater in the warmer regime.

For the characterization of these two regimes we assume a maximum temperature for non-fractioning sublimation in the model: $T_{\text{subl,max}}$. For $T_{\text{skin}} < T_{\text{subl,max}}$, we assume non-fractioning sublimation. In the case of $T_{\text{subl,max}} < T_{\text{skin}} < 0°$C, we assume equilibrium fractionation during the evaporation of meltwater.





To assess $T_{\text{subl,max}}$ for optimal agreement between modeled and observed $\delta D$, we performed 16 model runs with a different $T_{\text{subl,max}}$ in each run ($-15$ to $0°$C in steps of 1 K). We refer to these runs as $M_{\text{S\_MW},T_{\text{subl,max}}}$. The thick black line in Fig. 10 shows the mean differences between modeled and observed $\delta D$ of the 178 selected data points ($\Delta\delta D$) from the different $M_{\text{S\_MW},T_{\text{subl,max}}}$. Modeled $\delta D$ ratios are highest in $M_{\text{S\_MW},0°\text{C}}$, which assumes non-fractioning sublimation of snow

for all $T_{\text{skin}}<0°$C and is therefore identical with $M_{\text{S}}$. In $M_{\text{S\_MW},-15°\text{C}}$ the modeled $\delta D$ ratios are lowest. $\delta D$ ratios from $M_{\text{S\_MW},-15°\text{C}}$ are close to $\delta D$ from $M_{\text{MW}}$, as most $T_{\text{skin}}$ along the back trajectories were above $-15°$C, which means that almost no sublimation is considered. $M_{\text{S\_MW},-29°\text{C}}$ would give the same results as $M_{\text{MW}}$.

The agreement between observed and modeled $\delta D$ from the $M_{\text{S\_MW},T_{\text{subl,max}}}$ is best for a $T_{\text{subl,max}}$ of $-7.1°$C. For this $T_{\text{subl,max}}$ the mean $\Delta\delta D$ of all selected data points is 0 and also $\delta D$ of group "cold" as well as the $\delta D$ of group "warm" is

10 approximately reproduced by the model. Figure 9c shows the respective two-dimensional probability distributions.

The statistical uncertainty of this optimal $T_{\text{subl,max}}$ due to scatter between modeled and observed $\delta D$ is $0.7°$C. Further uncertainty is introduced to the determined optimal $T_{\text{subl,max}}$ by the assumptions of the Lagrangian isotope model. To assess this uncertainty of $T_{\text{subl,max}}$, we varied side constraints ($_{\text{type\_of\_variation}}$) in the $M_{\text{S\_MW},T_{\text{subl,max}}}$ analogous to the uncertainty assessment for the average modeled $\delta D$. For this purpose we performed $16·8=128$ model runs (Fig. 10, thin lines), which we

refer to as $M_{\text{S\_MW,type\_of\_variation},T_{\text{subl,max}}}$. Model runs with assumptions related to higher modeled $\delta D$ ratios (dashed thin lines) result in a lower optimal $T_{\text{subl,max}}$ and model runs with assumptions related to lower modeled $\delta D$ ratios (solid thin lines) result in a higher optimal $T_{\text{subl,max}}$.

Thin black lines in Fig. 10 depict the maximum possible shift of the average modeled $\delta D$ in Karlsruhe in the case of superposition of the three examined side constraints. The solid thin black line reflects a maximum unfavorable superposition of

20 assumptions related to lower modeled $\delta D$ ratios ($M_{\text{S\_MW},---,T_{\text{subl,max}}}$). The $M_{\text{S\_MW},---,T_{\text{subl,max}}}$ therefore allow to assess the upper bound of $-3.2°$C of the $T_{\text{subl,max}}$ for optimal agreement between model and observation. The lower bound of the optimal $T_{\text{subl,max}}$, which is derived from the $M_{\text{S\_MW},+++,T_{\text{subl,max}}}$, is $-15°$C. Here again, individual analysis of data from the groups "cold" and "warm" allows us to refine the result. For $M_{\text{MW},+++}$ $\Delta\delta D_{\text{warm}}$ is $+9.3$‰ (Table 1). Since $M_{\text{MW},+++}$ marks the lower boundary of $\delta D$ ratios from the $M_{\text{S\_MW},+++,T_{\text{subl,max}}}$, the corresponding set of assumptions doesn't al-

25 low reproducing the observations in group "warm" even if assuming a very low $T_{\text{subl,max}}$. Assuming maximum unfavorable superposition of the uncertainties of side constraints in the $M_{\text{S\_MW},+++,T_{\text{subl,max}}}$ is therefore too conservative for the uncertainty assessment of the optimal $T_{\text{subl,max}}$. For this reason, we assess the lower bound of the optimal $T_{\text{subl,max}}$ by means of the $M_{\text{S\_MW},+,\text{snow},T_{\text{subl,max}}}$, which only consider one uncertainty term and just allow the reproduction of the observed $\delta D$ in group "warm". From the $M_{\text{S\_MW},+,\text{snow},T_{\text{subl,max}}}$ we derive a better confined lower bound of the optimal $T_{\text{subl,max}}$ of $-10.8°$C.

30 So the uncertainty of model assumptions translates into an uncertainty range of $T_{\text{subl,max}}$ for optimal agreement of model and observation from $-10.8$ to $-3.2°$C. Together with the statistical uncertainty of $0.7°$C, the total uncertainty range of $T_{\text{subl,max}}$ sums up to $-11.5$ to $-2.5°$C.




## 6 Conclusions

In this paper, we investigated isotope fractionation during the sublimation or evaporation of snow at ground level. For this purpose, we combined 17 months of measurements of $\delta D$ in low-level water vapor in Central Europe with a new Lagrangian isotope model.

By means of this approach, we identified two regimes of GDAS skin temperatures ($T_{\text{skin}}$) below the freezing point with significantly different deviation between modeled and observed $\delta D$. To resolve this difference, we suggest two regimes of $T_{\text{skin}}$ with different types of predominant isotope fractionation. Based on sensitivity tests with the Lagrangian isotope model, we found that the colder regime is described best by non-fractioning sublimation of snow. The warmer regime is described best by fractioning evaporation of meltwater.

We determined a $T_{\text{subl,max}}$ separating both regimes of $T_{\text{skin}}$ by optimizing the agreement between modeled and observed $\delta D$. For $T_{\text{subl,max}}$ of $-7.1°$C this agreement is best. Uncertainty related to assumptions of the isotope model corresponds to a range of uncertainty of $T_{\text{subl,max}}$ from $-11.5$ to $-2.5°$C.

The finding of a cold temperature regime with a small impact of fractionation during sublimation at ground level does not contradict earlier studies on snow which indicate fractioning interaction between the snowpack and atmospheric water vapor, even in cases of temperatures far below the freezing point (Epstein et al., 1965; Stichler et al., 2001). These studies document systematic post-depositional increases of isotope ratios in the snowpack, which imply processes such as slight kinetic fractionation during sublimation in consequence of different coefficients of diffusion of the different water isotopes or fractioning vapor deposition. Given the uncertainty of the Lagrangian isotope model, these small effects would not be detectable by our approach. Nevertheless, these effects might result in significant post-depositional modifications of isotope ratios in the snowpack on timescales longer than the five days covered by the trajectories.

For GDAS skin temperatures between $T_{\text{subl,max}}$ and $0°$C our results imply significant fractioning evaporation of meltwater. Despite skin temperatures below the freezing point, the formation of meltwater is likely to exist within this temperature regime. Since we weighted skin temperatures with positive latent heat flux at ground level, $T_{\text{subl,max}}$ already refers to skin temperatures during the day, when evaporation is strongest. However, due to the coarse resolution of the six-hourly 1×1° GDAS data, locations of locally or temporally enhanced skin temperatures are smoothed out. The evaporation of meltwater at these locations may exceed the amount of moisture from sublimation for skin temperatures above $T_{\text{subl,max}}$. The identification of a fractioning "meltwater regime" is consistent with earlier observations of isotope ratios in snow, which point to fractioning evaporation during ablation at temperatures close below the freezing point (Moser and Stichler, 1974; Gurney and Lawrence, 2004; Lechler and Niemi, 2011). Since snow samples give an integrated signal over long time periods, a detailed attribution of these observations to certain meteorological conditions is difficult. Complementary to the studies on snow, the method presented here allows the post-depositional isotope fractionation to be attributed to meteorological conditions with GDAS skin temperatures between $T_{\text{subl,max}}$ and $0°$C.

Our results show that surface evaporation in the two identified regimes of skin temperature has a strong impact on the $\delta D$ of low-level water vapor in Central Europe. For isotope applications based on relations between $\delta D$ and temperature, the con-



sideration of the different types of isotope fractionation in both regimes is of great interest. For instance, seasonal ablation in coastal regions of Greenland might systematically affect the relation between the $\delta D$ of water vapor and dew point temperature over Greenland. Because such a seasonality may be different in climatologically different time periods, it may introduce uncertainty to temperature reconstructions from Greenland ice cores.

5  Furthermore, fractioning evaporation of meltwater will increase the $\delta D$ ratio of the residual meltwater and in the case of recrystallization, the $\delta D$ ratio of the snowpack. Ignoring fractioning evaporation therefore introduces uncertainty to a variety of isotope applications from reconstructions of paleotemperatures and paleotopography to studies on the formation of groundwater. The specification of a temperature regime with enhanced fractionation during evaporation may therefore help to identify, investigate, and reduce biases inherent to these applications.

10  *Acknowledgements.* This study was funded in part by the European Research Council under the European Community's Seventh Framework Programme (FP7/2007-2013)/ERC grant agreement no. 256961 and by the German Climate Modeling Initiative (PalMod). We acknowledge support by Deutsche Forschungsgemeinschaft and Open Access Publishing Fund of Karlsruhe Institute of Technology.



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

1)





**Table 1.** Average differences between modeled and measured $\delta D$ of the 178 selected data points ($\Delta\delta D$), data points of group "cold" ($\Delta\delta D_{\mathrm{cold}}$), and data points of group "warm" ($\Delta\delta D_{\mathrm{warm}}$) from different model runs ($M$). Values of particular interest are printed in bold type.

| Name of model run | Description of model run | $\Delta\delta D$ | $\Delta\delta D_{\mathrm{cold}}$ | $\Delta\delta D_{\mathrm{warm}}$ |
|---|---|---|---|---|
| $M_{\mathrm{MW}}$ | fractioning evaporation of meltwater; (**reference run**) | **−19.7‰** | −28.3‰ | −11.0‰ |
| $M_{\mathrm{MW+,snow}}$ | fractioning evaporation of meltwater; $\delta D$ ratios of snow increased by 11.5‰ | −10.5‰ | −19.0‰ | −2.0‰ |
| $M_{\mathrm{MW+,ini}}$ | fractioning evaporation of meltwater; $\delta D$ ratios at initialization increased by 11.8‰ / 100‰ for air masses originating from altitudes lower / higher than 2000 m above ground level | −13.4‰ | −23.1‰ | −3.8‰ |
| $M_{\mathrm{MW+,upt.36h}}$ | fractioning evaporation of meltwater; reduced moisture uptake in consequence of smoothing $q$ along the trajectories with a 36 h broad rectangle kernel (instead of 24 h) | −17.3‰ | −26.9‰ | −7.7‰ |
| $M_{\mathrm{MW+++}}$ | fractioning evaporation of meltwater; simultaneous occurrence of the three assumptions above | −1.3‰ | **−11.9‰** | **+9.3‰** |
| | | | | |
| $M_{\mathrm{S}}$ | non-fractioning sublimation of snow | **+25.7‰** | +22.7‰ | +28.6‰ |
| $M_{\mathrm{S−,snow}}$ | non-fractioning sublimation of snow; $\delta D$ ratios of snow decreased by 11.5‰ | +15.9‰ | +12.8‰ | +19.0‰ |
| $M_{\mathrm{S−,ini}}$ | non-fractioning sublimation of snow; $\delta D$ ratios at initialization dencreased by 3.6‰ / 100‰ for air masses originating from altitudes lower / higher than 2000 m above ground level | +23.2‰ | +21.0‰ | +25.4‰ |
| $M_{\mathrm{S−,upt.12h}}$ | non-fractioning sublimation of snow; increased moisture uptake in consequence of smoothing $q$ along the trajectories with a 12 h broad rectangle kernel (instead of 24 h) | +24.8‰ | +21.1‰ | +28.4‰ |
| $M_{\mathrm{S−−−}}$ | non-fractioning sublimation of snow; simultaneous occurrence of the three assumptions above | **+13.5‰** | +10.3‰ | +16.7‰ |





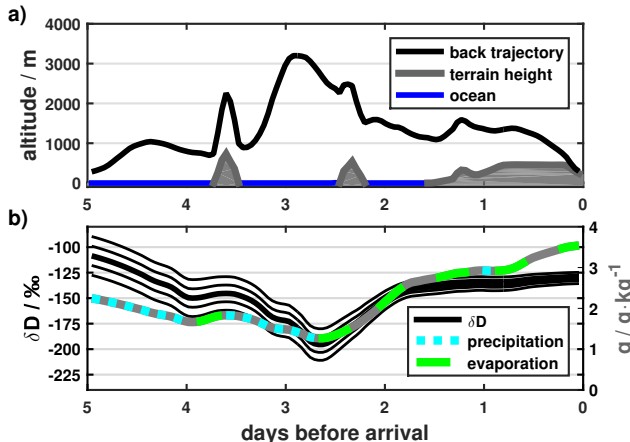

**Figure 1.** Illustration of the isotope modeling for one exemplary back trajectory (arrival in Karlsruhe March 18, 2012, 0 UTC). **(a)** Altitude of the back trajectory (black) and terrain height (gray/blue). The isotope model was initialized at $80°$ N in the marine boundary layer (MBL). In a low-pressure system near Iceland the tracked air parcel ascended to an altitude of $3200\,m$. During the last three days of transport to Karlsruhe the air parcel was sinking to the sampling altitude. **(b)** After initialization in the MBL, $\delta D$ of the tracked air parcel (thick black line) was slightly decreasing, due to the formation of precipitation (dashed blue lines) within the first day. More pronounced formation of precipitation, in consequence of lofting in a low-pressure system near Iceland, resulted in a second pronounced decrease of specific humidity $q$ (gray) and the modeled $\delta D$ ratio dropped accordingly. Due to moisture uptake (dashed green lines), related to a descent of the air parcel in the subsequent days, $q$ and the $\delta D$ ratio were increasing, until the air parcel reached Karlsruhe. Black thin curves illustrate the modeled $\delta D$ for different initializations of $\delta D$ (section 2.3.3). The dependence on the initialization decreases with the amount of moisture uptake along the trajectories and is only low in Karlsruhe.





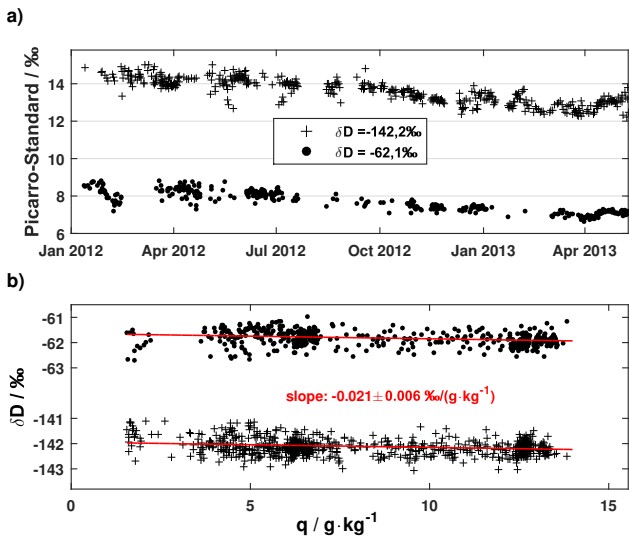

**Figure 2. (a)** Deviation of individual calibration measurements with the Picarro water isotopologue analyzer from isotope ratios of the liquid standards. Each point represents a calibration of one hour. Dots: Standard 1; crosses: Standard 2. **(b)** Humidity dependence of the $\delta D$ measurements. Long-term drift depicted in (a) was removed in (b) by subtracting the five-week running average of calibrations. Red regression lines were calculated for both standards simultaneously by subtracting the mean difference between both standards. $\pm$ gives the difference between the slopes calculated for the first and second half of the measurement period.





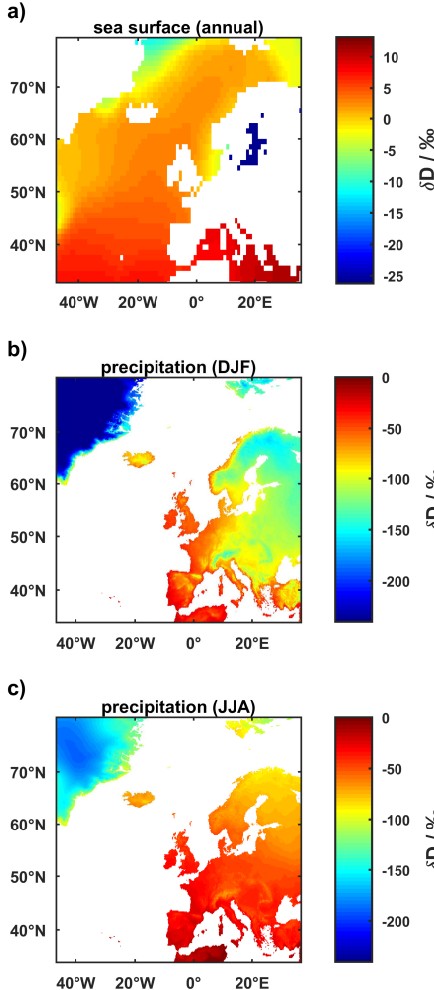

**Figure 3. (a)** $\delta D$ of ocean surface water derived from the interpolation (V1.1) of the Global Seawater Oxygen-18 Database by LeGrande and Schmidt (2006), assuming a constant factor of 6.56 between $\delta D$ and $\delta^{18}O$ in ocean surface water. **(b)** Climatological $\delta D$ in precipitation in winter (DJF) from the Regionalized Cluster-based Water Isotope Prediction (RCWIP), which in turn is based on observations of the Global Network of Isotopes in Precipitation (GNIP). **(c)** same as in (b) but for summer (JJA).





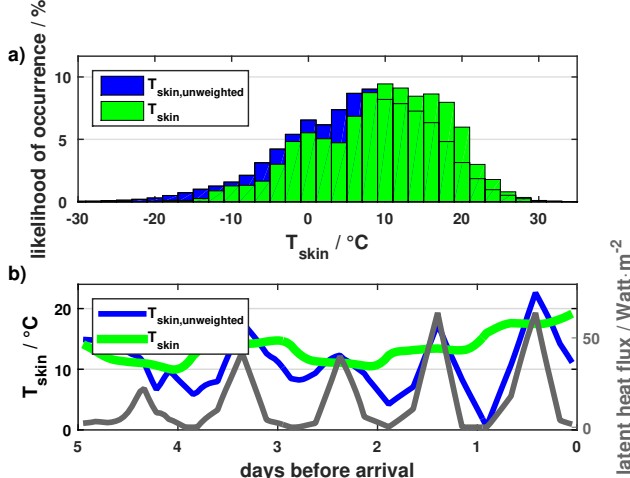

**Figure 4.** (a) Probability distributions of continental GDAS skin temperatures ($T_{skin,unweighted}$) (blue) and of skin temperatures weighted with the accumulated hourly latent heat flux at ground level within $\pm 12\,h$ ($T_{skin}$) (green). The occurrence of low temperatures is reduced as a consequence of the weighting. A peak around $0°C$ becomes more clearly visible. (b) Illustration of the weighting algorithm for one exemplary back trajectory (arrival in Karlsruhe May 4, 2012, 21 UTC).

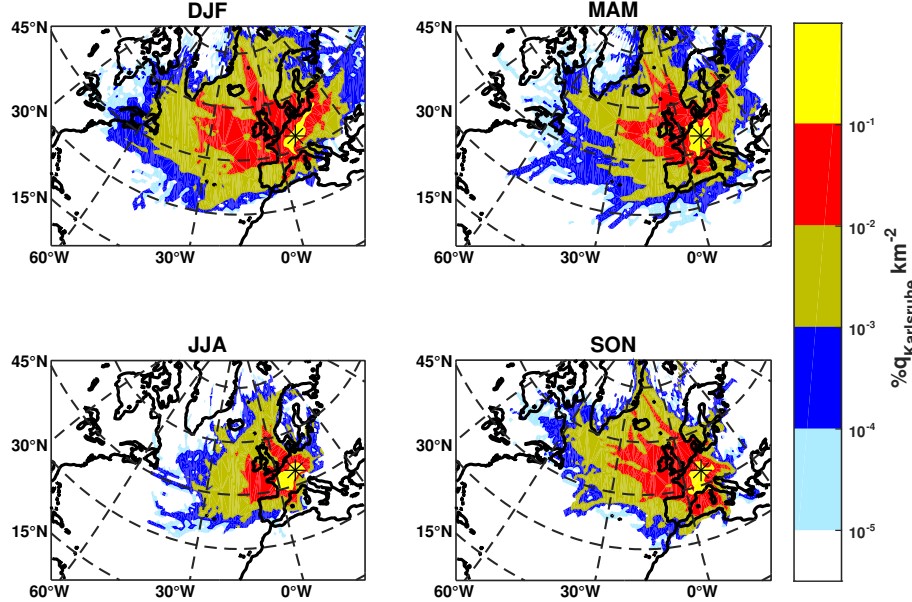

**Figure 5.** Source regions of moisture ($q$) $30\,m$ above ground level in Karlsruhe (black star) in the different seasons: December, January, February (DJF), March, April, May (MAM), June, July, August (JJA), September, October, November (SON). The color code indicates the contribution of different source regions to $q$ in Karlsruhe in % per $km^2$. Integration over the whole map gives the identified humidity per season. Because of the finite length of the back trajectories, the total identified humidity is lower than 100% and accounts for 52% (DJF), 51% (MAM), 39% (JJA) and 50% (SON).





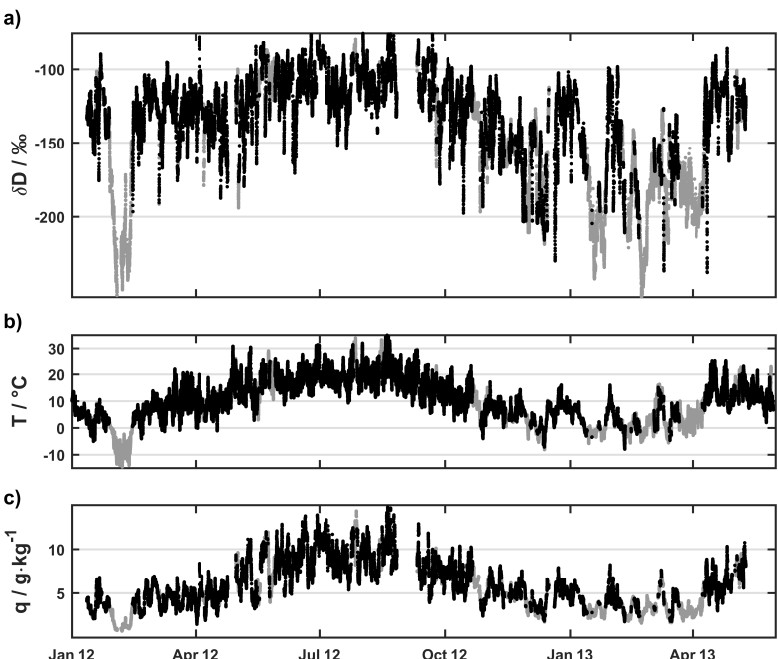

**Figure 6.** Measurements in Karlsruhe from January 2012 until May 2013 30 m above ground level (ten-minute averages). Black: five-day back trajectories originate from the West; gray: five-day back trajectories originate from the East. **(a)** $\delta D$ of water vapor. Gaps in the time series are caused by instrumental issues with analyzer and calibration device. **(b)** Air temperature ($T$). **(c)** Specific humidity ($q$).





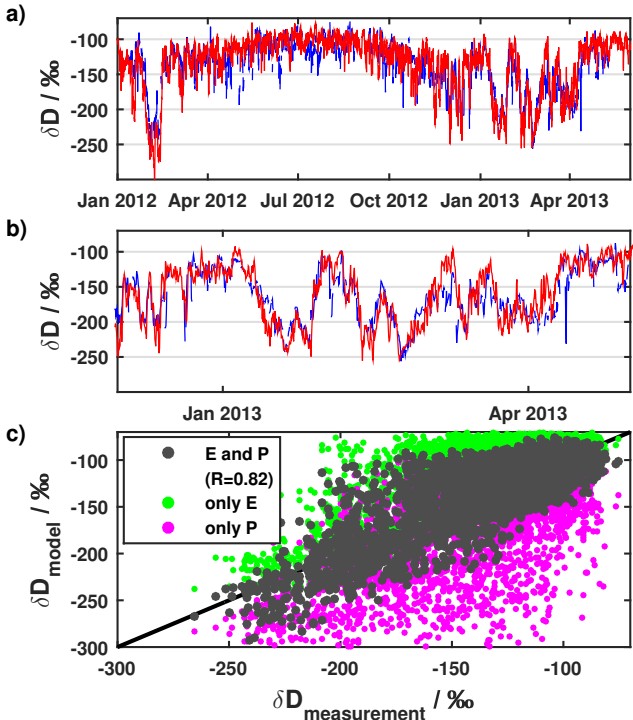

**Figure 7.** **(a)** Time series of measured (blue) and modeled (red) $\delta D$ of water vapor in Karlsruhe. **(b)** Enlarged section of (a), which demonstrates the capability of the model of capturing the high variability of $\delta D$ in winter. **(c)** Measured versus modeled $\delta D$. The three-hourly available modeled $\delta D$ is compared to the temporally closest ten-minute average of the $\delta D$ observations. Gray: reference run, surface evaporation (E) and the formation of precipitation (P) are considered (R=0.82, $\Delta\delta D$=+1.0‰); green: only E is considered, P is ignored (R=0.81, $\Delta\delta D$=+29.4‰); magenta: only P is considered, E is ignored (R=0.62, $\Delta\delta D$=−35.6‰); black: 1:1 line.

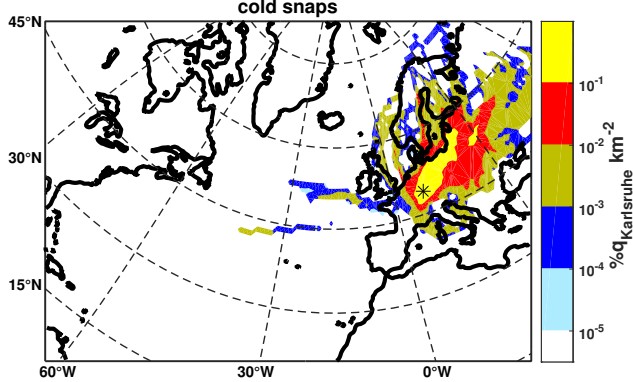

**Figure 8.** Lagrangian source region analysis of low-level water vapor in Karlsruhe for the observed cold snaps. The mean identified fraction of moisture along the five-day back trajectories is 48%.





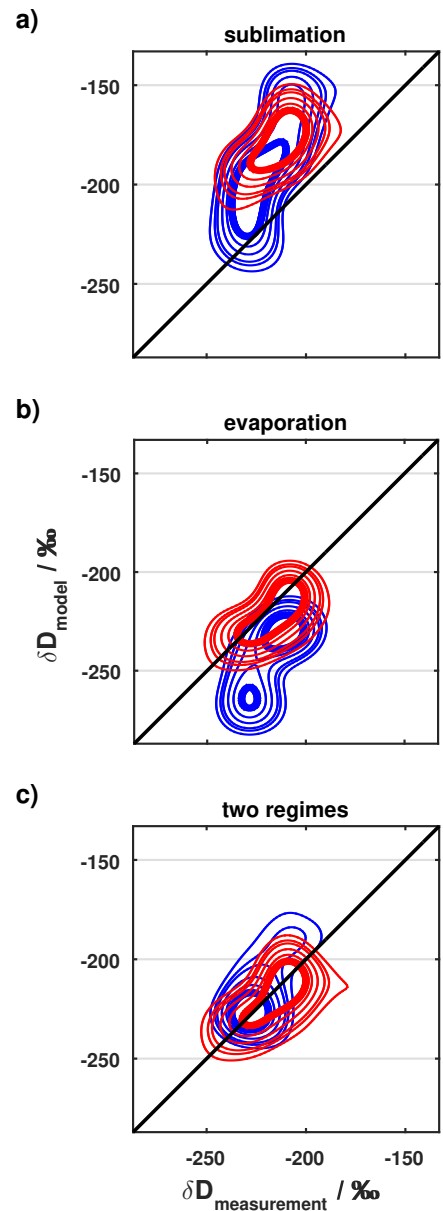

**Figure 9.** Two-dimensional probability distributions of measured and modeled $\delta D$ of low-level water vapor in Karlsruhe for selected cold snap events. Red: group "warm"; blue: group "cold". Probabilities were calculated for a 20‰×20‰ $\delta D$ grid, smoothed with a 20‰ broad rectangle kernel, and finally interpolated to a 1‰×1‰ grid. Probabilities are normalized to 1 at the maximum, contours show probability levels of **0.8**, 0.7, 0.6, 0.45, 0.35. **(a)** The model assumes sublimation of snow (no isotope fractionation) in the case of moisture uptake and skin temperature ($T_{\mathrm{skin}}$) below 0°C. **(b)** The model assumes evaporation of melted snow (equilibrium isotope fractionation) in the case of moisture uptake and $T_{\mathrm{skin}}<0$°C. **(c)** The model assumes sublimation of snow in the case of moisture uptake and $T_{\mathrm{skin}}<-7.1$°C. In the case of moisture uptake and $-7.1$°C$<T_{\mathrm{skin}}<0$°C the model assumes evaporation of melted snow.




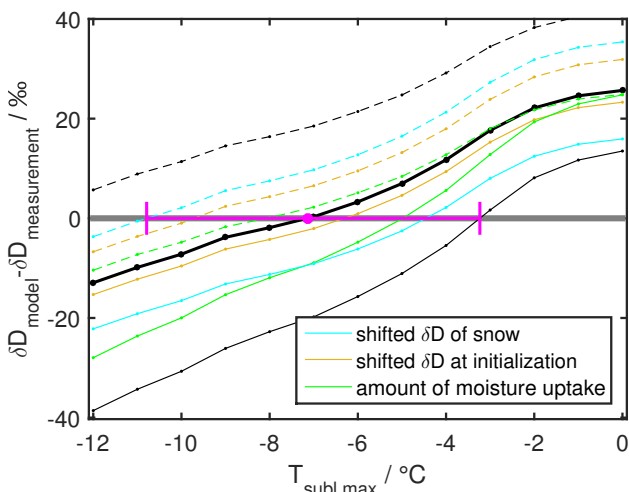

**Figure 10.** Mean difference between modeled and observed $\delta D$ of water vapor at Karlsruhe for selected cold snap events ($\Delta \delta D$, thick black line). The modeled $\delta D$ ratios increase with increasing maximum skin temperature allowing non-fractioning sublimation ($T_{\mathrm{subl,max}}$). Magenta: optimal $T_{\mathrm{subl,max}}$ for best agreement of model and observation (dot) and uncertainty of the optimal $T_{\mathrm{subl,max}}$ (error bar) due to potential systematic deviation of $\delta D$ of snow to climatology (blue), initialization of $\delta D$ (yellow), and uncertainty of the amount of moisture uptake (green). Thin black lines: modeled $\delta D$ in the case of superposition of the different sources of uncertainty.