# Peer review of "The influence of snow sublimation and meltwater evaporation on $\delta D$ of water vapor in the atmospheric boundary layer of Central Europe"

_Atmospheric Chemistry and Physics, 2016_

## Referee Comment (RC1) · Anonymous Referee #1 · 27 Jul 2016

Review of

**"The influence of snow sublimation on stable isotopes of water vapor in the atmospheric boundary layer of Central Europe"**

by E. Christner et al.

Paper published in ACPD on 13 June 2016

**1   General Comments**

In this paper continuous measurements of $\delta$D in near-surface atmospheric vapour from a station near Karlsruhe in Germany are presented and used together with a Lagrangian isotope model to investigate the moisture source conditions during cold spells. The authors find that below a critical skin temperature of $-7.1^{\circ}$C the moisture sources are dominated by non-fractionating sublimation, whereas above this critical skin temperature and up to $0^{\circ}$C fractionating evaporation of meltwater is dominating. The separation into the two fractionation regimes is done by comparing the modelled $\delta$D with the measurements in Karlsruhe. These results have important implications for the isotope modelling of atmospheric moisture sources in snow covered regions, which are generally treated in a very simplistic way in isotope-enabled numerical models (only non-fractionating sublimation, see e.g. Yoshimura et al. (2006); Werner et al. (2016)).

I recommend publication of this original, very well-written paper and I suggest only a few minor revisions that are listed below.

**2   Specific comments**

1. p. 1, Title: In my opinion it would be helpful to refine the title. First, I think it might be interesting to state somewhere in the paper (maybe in Section 4) the percentage of the total moisture measured over the 17 months in Karlsruhe that originates from snow covered regions. This would quantify the importance of snow covered moisture sources for Central Europe. Second, I wondered whether adding "meltwater evaporation" to have "The influence of snow sublimation and meltwater evaporation..." would be helpful for a potential reader. Third, I think it would be good to say "... on $\delta$D of water vapour...".

2. p. 1, L. 3: Is "evaporation of snow" really what the authors mean? Isn't it sublimation of snow and evaporation of meltwater to be precise? So could one say maybe "isotope fractionation at snow covered moisture sources" or something similar to stay general and not imply only one of the two fractionation regimes that are discussed?

3. Abstract in general: 1) It might be helpful for the reader to see the suggested quantified importance of snow covered moisture sources (percentage of snow covered moisture sources in the measurement period) as suggested in the first comment also in the abstract. The 17 months of data when reading about snow sublimation and meltwater evaporation seems a bit surprising. 2) I would suggest one last sentence to conclude the abstract with an outlook on the impact of the major finding presented in this study on modelling and experimental stable water isotope research. Ideally saying that the existence of two fractionation regimes has important implications for the more realistic modelling of isotope processes at snow covered moisture sources and that more detailed experimental studies at snow covered sites are needed to better describe the potential coexistence of the two regimes.

4. The introduction was a pleasure to read!

5. p. 3, L. 15: A short discussion on the choice of the length of the back-trajectories and on the potential reduction of uncertainty by using e.g. 10 days back trajectories should be provided. Läderach and Sodemann (2016) use the same moisture source diagnostics as the authors of this paper and find about 4–5 days for the global mean residence time.

6. p. 3, L. 22: Additional vertical displacement of the trajectories would allow an assessment of the uncertainty related to the arrival altitude of the investigated air parcels.

7. p. 4, L. 30: Does "$q$-weighted" mean weighted with $q$@arrival? If so it would be helpful to say it explicitely.

8. p. 5, L. 10-15: In the considered cold snap conditions, I am not totally convinced that supersaturation in ice clouds would be so low. Is there a way to assess this source of uncertainty for the subsequent analysis of snow covered moisture sources?

9. p. 5, L. 13: I would recommend to say "examined transport conditions" or "examined moisture transport regions" or similar.

10. p. 5, L. 22-25: I am not sure if I understand this paragraph correctly but it points to an important assumption and source of uncertainty of the moisture source diagnostics. Could the authors rephrase? Maybe just removing "In this case" and adding at the end of the last sentence ", only by the freshly evaporated moisture"?

11. p. 5, L. 30: Could the authors quantify the influence of non-equilibrium fractionation on $\delta$D? It should be around 1-3‰ depeding on the environmental conditions.

12. p. 7, L. 1: No seasonal changes in FT?

13. In general: I would suggest not using italics for $^{18}$O and D.

14. p. 8, L. 8: "ignored"? What does that mean? That it does not impact total uncertainty?

15. p. 9, L. 8: "of the whole calibrations", maybe clearer if one said "found using all calibrations".

16. p. 9, L. 13: What does "WSW" mean?

17. p. 10, L. 12: It would be nice to have the period that is covered by the measurements right at the beginning of Section 4.

18. p. 11, L. 4: Fig 3c in JJA I do not see the continental effect so clearly, is it just the colorbar range or is it that in summer continental recycling smears out the strong continental gradient visible in winter. So should only a reference to Fig. 3b be given here?

19. p. 12, L. 7: Shouldn't it say "Sublimation of snow or snow-melt evaporation?"?

20. p. 12, L. 19: "To this end, we identified moisture uptake at $T_{skin} < 0$°C". This is a repetition the sentence before is enough in my opinion.

21. p. 14, L. 8: I assume the air masses originating from higher altitudes are also very dry so that the subsequent history of the air mass plays an important role and particularly subsequent uptakes so that I would not be so worried about the uncertainty associated with these air parcels' initialisation.

22. p. 15, L. 18: In table 1 only $\Delta\delta$D are stated and I would find it important to also mention the root mean square or absolute difference. Figure 9 of course also helps with respect to this point.

23. p. 16, L. 8: °$C$ should be C

24. p. 24, Figure 1: I find this Figure very helpful. I just did not understand whether the dashed blue and green lines indicate exact locations of precipitation/uptakes? If yes then why is the green line not continuous between -2.5 and -2 days? If no, then the authors should explain how the lines should be interpreted in the caption.

25. p. 27, Figure 5: state the period associated with these climatological source regions.

26. p. 29, Figure 8: Why is the total identified moisture along the trajectories only 48%? It seems low to me. Is it due to the trajectory length?

**References**

Läderach, A., and H. Sodemann: A revised picture of the atmospheric moisture residence time, *Geophys. Res. Lett.*, 43, 924–933, doi:10.1002/2015GL067449, 2016.

Werner, M., Haese, B., Xu, X., Zhang, X., Butzin, M., and Lohmann, G.: Glacial–interglacial changes in H218O, HDO and deuterium excess – results from the fully coupled ECHAM5/MPI-OM Earth system model, *Geosci. Model Dev.*, 9, 647–670, doi:10.5194/gmd-9-647-2016, 2016.

Yoshimura, K., Miyazaki, S., Kanae, S., and Oki, T.: Iso-MATSIRO, a land surface model that incorporates stable water isotopes, *Global and Planetary Change*, 51, 90-107, doi:10.1016/j.gloplacha.2005.12.007, 2006.

---

## Referee Comment (RC2) · Anonymous Referee #2 · 21 Nov 2016

Review for
*The influence of snow sublimation on stable isotopes of water vapor in the atmospheric boundary layer of Central Europe*

**General comments:**

The manuscript by Christner et al. presents 17 months of new, continuous, low-level atmospheric water vapor $\delta D$ measurements at a site near Karlsruhe, Germany in conjunction with a Lagrangian isotope model to inform understanding of the along-trajectory controls on measured $\delta D$ variability. Integration of HYSPLIT-calculated, low-level back-trajectories with the isotope model shows that much of the observed variability in measured $\delta D$ values is the result of identifiable along-transport processes most generally associated with moistening and dehydration of the air parcel through evapotranspiration and precipitation processes, respectively. A subset of back-trajectories associated with 'cold snaps' result from a systemic shift to continental source regions and easterly trajectories. For this subset, the authors investigate an additional controlling mechanism for isotopic evolution of these easterly trajectories, that of isotopic modification via moistening from surface snow sublimation. In the closing section of the manuscript, the authors investigate a range of possible conditions for isotopic modification of regional snowpacks that best explains measured $\delta D$, namely skin temperature controls on fractionating versus non-fractionating sublimation processes. From this, the authors determine the relevant skin temperature window for which post-depositional isotopic modification of snowpacks and associated impacts on low-level atmospheric moisture $\delta D$ is most relevant.

This is a detailed and comprehensive manuscript that presents a new and robust long-term $\delta D$ dataset that proves valuable for investigating controls on the isotopic evolution of low-level atmospheric moisture. The methods applied are appropriate. I particularly like the integration of HYSPLIT-derived trajectories into the new Lagrangian model presented and find the observed-model $\delta D$ congruence (Figs. 6 and 7) impressive and supporting of the Lagrangian-model approach. I find the discussion of the isotope effects of sublimation both nuanced and convincing, which is important given that this impactful process is generally neglected or assumed to be negligible in similarly-focused studies. I expect the findings and research design of this study will be of interest to a broad audience, particularly in light of the expansion of laser-based isotope analyzers that are sure to increase the number of similar isotope records in coming years. Accordingly, I strongly support publication of the manuscript in ACP pending some minor/moderate revisions to the text and some figures in order to (1) reduce redundancies in the text, (2) provide additional clarification for components of the HYSPLIT-Lagrangian isotope model integration and (3) ensure the more complex aspects of the manuscript are understandable to the broader audience likely to be interested in this work (e.g., earth scientists interested in proxy-based investigations of paleoaltimetry and paleoclimate).

**Specific comments:**

(1) *Reorganization and reducing redundancies* – much of the text in Section 3.3 is redundant with Section 2.3.2. It seems much of 3.3 could be moved and combined with 2.3.2. Similarly, section 3.4 falls under the heading 'Measurements'; however, this moisture source data seems more relevant to the model description in Section 2.3.

The opening paragraph of Section 2.3.3 presents some basic back-trajectory statistics but explanation is limited here. Important clarifying information is not provided until Section 4, specifically Section 4.3. I suggest saving back-trajectory statistics for Section 4 when more details needed for clarification are presented.

Discussion of 24-hour smoothing window on page 4 could be removed there and saved for the same discussion on p. 14 (lines 19-28).

(2) *Model clarification and limitations* – Given the uncertainty in vapor δD at high altitudes (> 2 km) and low proportion of trajectories (2%) encountering these altitudes, it seems these trajectories could simply be removed from the analysis.

Beyond the vapor measurements at the Karlsruhe study site, are there other published datasets that would ground truth the isotope model presented? Specifically, are there any regional records of soil water δD (δ$^{18}$O) that can be presented to constrain the RCWIP values used? Additionally, are there any regional snowpack δD (δ$^{18}$O) records that would give a better understanding of the degree of snowpack δD variability? It is likely that snowpack δD varies both spatially and temporally throughout the accumulation and melt season in the study region, thus some discussion on how this variability limits the model presented is important.

Another limitation that might be more explicitly discussed or tied into the previous point about spatial δD variability of the snowpack is that of the 1°x1° resolution of the GDAS data set. How might this spatial smoothing impact ability to model δD variation?

(3) *Clarifying complexity* –I find Section 5.2 and corresponding Figure 10 difficult to comprehend. I understand the general idea that multiple model runs were used to identify the cutoff temperature between fractionation and non-fractionating sublimation, but is not clear to me how the 16 scenarios of 'side constraint' variability and associated 128 total model runs correspond to the lines shown in Fig. 10. How do 128 total model runs translate to 9 distinct lines in Fig. 10? Please clarify in the text and figure caption and reconsider what Figure 10 should show to more clearly communicate the information in this section.

Given the focus on easterly trajectories, a figure more clearly showing association of easterly trajectory pathways with corresponding snow cover in that region would be helpful. This might be accomplished by adding a panel to Figure 8 showing snow cover.

(4) The title, introduction, and conclusions sections place focus exclusively on the sublimation aspects of this work. I think this undersells and undervalues the importance of the broader Lagrangian isotope model approach and its applications. Sublimation appears to only relate to the easterly trajectory subset (< 50% of trajectories investigated). I am not sure if there is a companion paper planned/submitted/published for the (north)westerly trajectories but these trajectories seem important to discuss in more detail as well, even with a single summary paragraph somewhere in Section 4 of what was learned from modeling these trajectories. If manuscript focus is to be exclusively on sublimation effects, as the title implies, I don't think all trajectory data should be included (i.e., Fig. 5) and the δD record should focus on 'cold snap' trajectories. Currently, there seems to be too much data shown in Figures 5-7 that receive too cursory of a discussion.

Review for
*The influence of snow sublimation on stable isotopes of water vapor in the atmospheric boundary layer of Central Europe*

**Technical comments**

*p. 1, Line 16:* 'isotopologues' of water more appropriate since molecules are listed

*p. 2, Lin4 4:* somewhat misleading to say snowpacks are most sensitive to isotopic modification. Lakes, for example, generally provide longer-duration integration of post-depositional modification.

*p. 2, Lines 7-8:* Here and throughout I think 'fractioning' should be replaced with 'fractionating'. I am unfamiliar with the term 'fractioning' with regards to isotopes.

*p. 2, Line 8:* clarify what is meant by 'other part'

*p. 2, Line 13:* It might be helpful to introduce the delta notation earlier in the manuscript so that it can be used here, i.e., 'increase of δ values'

*p. 2, Line 21:* What is meant by 'resublimation'?

*p. 4, Lines 12-13:* Quantify what is meant by 'fast' – sub-diurnal?

*p. 4, Line 15:* What is meant by 'displacement' of moisture uptake amount?

*p. 5, Line 3:* I think the phrasing 'more likely' should be changed – 'preferential fractionation of D into the liquid phase' is more precise language.

*p. 5, Line 25:* Clarify what is meant by 'not affected by dilution'

*p. 6, Line 15:* Where does the 2.6 day value come from?

*p. 11, Line 1:* Here and elsewhere, should use 'value' instead of 'ratio' to describe δD

*p. 11, Line 4:* I think the statement 'caused by the relation between δD and condensation temperature' is misleading. The more direct control on the continental effect is degree of rainout.

*p. 12, Lines 18-19:* Redundant sentences: 'Second….' And 'To this end….'

*p. 12, Lines 20-21:* Rationale for only considering trajectories above 28% median level needs to be more clearly communicated.

*p. 13, Lines 4-5:* The term 'side constraints' here and following is unclear to me. Can you clarify what is meant by 'side'?

*p. 15, Line 29:* Clarify what 'both groups' refer to.

---

## Author Comment (AC1) · 4 Jan 2017

Replies to reviewer 1

by E. Christner, M. Kohler, M. Schneider

January 4, 2017

We like to thank reviewer 1 for the review and the constructive comments, which helped to improve the manuscript.

According to the comments of reviewer 1, we implemented (1) kinetic fractionation during surface evaporation at skin temperatures above 0°C and (2) reduced fractionation under conditions of supersaturation in clouds into the model. Because of this, most numbers and figures throughout the manuscript are slightly changed compared to the initial version. A marked-up manuscript version is attached at the end of this document.

**General Comments**

In this paper continuous measurements of δD in near-surface atmospheric vapour from a station near Karlsruhe in Germany are presented and used together with a Lagrangian isotope model to investigate the moisture source conditions during cold spells. The authors find that below a critical skin temperature of -7.1°C the moisture sources are dominated by non-fractionating sublimation, whereas above this critical skin temperature and up to 0°C fractionating evaporation of meltwater is dominating. The separation into the two fractionation regimes is done by comparing the modelled δD with the measurements in Karlsruhe. These results have important implications for the isotope modelling of atmospheric moisture sources in snow covered regions, which are generally treated in a very simplistic way in isotope-enabled numerical models (only non-fractionating sublimation, see e.g. Yoshimura et al. (2006); Werner et al. (2016)).

Our reply: We added a reference to Yoshimura et al. (2006) and Werner et al. (2016) to the introduction: *"This uncertainty complicates interpretation of isotope records from snow covered regions, as the contribution of the various potential post-depositional effects may be different during climatologically different time periods. In addition, this uncertainty limits the isotope modeling of atmospheric moisture sources in snow covered regions, as isotope-enabled models in general only consider one of the types of isotope fractionation for the sublimation of snow (e.g.( Yoshimura, Miyazaki, Kanae, & Oki, 2006; Werner et al., 2016))".*

I recommend publication of this original, very well-written paper and I suggest only a few minor revisions that are listed below.

**Specific comments**

1. p. 1, Title: In my opinion it would be helpful to refine the title. First, I think it might be interesting to state somewhere in the paper (maybe in Section 4) the percentage of the total moisture measured over the 17 months in Karlsruhe that originates from snow covered regions. This would quantify the importance of snow covered moisture sources for Central Europe.

Our reply: We added the following paragraph to Sect. 5.1: *"According to the Lagrangian moisture source diagnostic, only 5% of the humidity analyzed in Karlsruhe from January 2012 to May 2013 originated from surface evaporation at locations with a GDAS based snow depth greater than 0.5 cm. For this reason, we used data fulfilling three selection criteria for the further investigation of isotope fractionation during evaporation from snow covered surfaces."*

Second, I wondered whether adding "meltwater evaporation" to have "The influence of snow sublimation and meltwater evaporation..." would be helpful for a potential reader. Third, I think it would be good to say "...on δD of water vapour...".

Our reply: We changed the title to *"The influence of snow sublimation and meltwater evaporation on δD of water vapor in the atmospheric boundary layer of Central Europe".*

2. p. 1, L. 3: Is "evaporation of snow" really what the authors mean? Isn't it sublimation of snow and evaporation of meltwater to be precise? So could one say maybe "isotope fractionation at snow covered moisture sources" or something similar to stay general and not imply only one of the two fractionation regimes that are discussed?
Our reply: We changed the sentence to: "*In this study, we investigate isotope fractionation at snow covered moisture sources by combining 17 months of observations of isotope concentration ratios [$HD^{16}O$]/[$H_2^{16}O$] in low-level water vapor in Central Europe with a new Lagrangian isotope model.*"

3. Abstract in general: 1) It might be helpful for the reader to see the suggested quantified importance of snow covered moisture sources (percentage of snow covered moisture sources in the measurement period) as suggested in the first comment also in the abstract. The 17 months of data when reading about snow sublimation and meltwater evaporation seems a bit surprising.
Our reply: We added the following sentence to the abstract: "*Observations from 38 days were associated with cold snaps and moisture uptake in snow covered regions.*"

2) I would suggest one last sentence to conclude the abstract with an outlook on the impact of the major finding presented in this study on modelling and experimental stable water isotope research. Ideally saying that the existence of two fractionation regimes has important implications for the more realistic modelling of isotope processes at snow covered moisture sources and that more detailed experimental studies at snow covered sites are needed to better describe the potential coexistence of the two regimes.
Our reply: We added the following two sentences at the end of the abstract: "*The existence of the two fractionation regimes has important implications for the interpretation of isotope records from snow covered regions as well as for a more realistic modeling of isotope fractionation at snow covered moisture sources. For these reasons, more detailed experimental studies at snow covered sites are needed to better constrain the $T_{subl,max}$ and to further investigate isotope fractionation in the two regimes.*"
We further added a paragraph to Sect. 6 clarifying the meaning of $T_{subl,max}$ and potential coexistence of the two regimes in a 1x1° grid cell.

4. The introduction was a pleasure to read!
Thanks!

5. p. 3, L. 15: A short discussion on the choice of the length of the back-trajectories and on the potential reduction of uncertainty by using e.g. 10 days back trajectories should be provided. Läderach and Sodemann (2016) use the same moisture source diagnostics as the authors of this paper and find about 4-5 days for the global mean residence time.
Our reply: We added the following paragraph to Sect. 2.3.3: "*Considering typical atmospheric moisture residence times in the range of 4–8 days (Läderach & Sodemann, 2016; Trenberth, 1998), using ten-day back trajectories instead of five-day back trajectories should almost eliminate model uncertainty of the δD at Karlsruhe from uncertainty of $R_{D,ini}$. However, such long trajectories easily cover distances of 10,000km, which makes the modeled δD sensitive to potentially very different conditions in far-distant regions. This, in turn, makes uncertainty assessment of the modeled δD more complex. Using five-day back trajectories is therefore a trade-off between a reasonably small sensitivity of the modeled δD at Karlsruhe on $R_{D,ini}$ and concentrating the analysis to the North Atlantic and Eurasia. Whenever it is necessary for the interpretation of our results, we assess uncertainty of the modeled δD from the model initialization by changing $R_{D,ini}$ in different model runs.*"
We further added a sentence to Section to Sect. 4.1: "*Both numbers are in reasonable agreement with more extended studies implying global atmospheric moisture residence times in the range of 4–8 days (Läderach & Sodemann, 2016; Trenberth, 1998).*"

6. p. 3, L. 22: Additional vertical displacement of the trajectories would allow an assessment of the uncertainty related to the arrival altitude of the investigated air parcels.

Our reply: We agree that vertical displacement of the trajectories allows an assessment of uncertainty with respect to the origin of back trajectories from different arrival altitudes or conditions in the moisture source regions. Considering the general humidity-dependence and the resulting altitude-dependence of δD, it is however not trivial to combine modeled δD from different altitudes and to compare such a combined δD with the observations. If simply weighting the δD from different altitudes with specific humidity the resulting combined δD values are likely to be systematically lower than the δD values at ground level. Because (1) robustness of the used back trajectories is already demonstrated by the high correlation between the modeled and the observed δD and (2) the difficulty of combining and comparing the δD from different altitudes, we don't use a "multi-altitude" trajectory ensemble in this paper.

7. p. 4, L. 30: Does "q-weighted" mean weighted with q@arrival? If so it would be helpful to say it explicitly.

Our reply: We changed the sentence to *"From each trajectory ensemble, nine modeled δD values for Karlsruhe are obtained, which are combined to one average value by weighting the nine values with q at the arrival."*

8. p. 5, L. 10-15: In the considered cold snap conditions, I am not totally convinced that supersaturation in ice clouds would be so low. Is there a way to assess this source of uncertainty for the subsequent analysis of snow covered moisture sources?

Our reply: We now included supersaturation in clouds into the model. This increased the mean δD of the groups cold / warm by 1.7 / 0.6 ‰. We replaced the sentence "…supersaturation in ice clouds (Merlivat and Jouzel, 1979; Jouzel andMerlivat, 1984) is ignored…" by *"A reduction of the fractionation factor $α_D(T)$ in the case of supersaturation in ice clouds (Merlivat and Jouzel, 1979; Jouzel and Merlivat, 1984) by a factor on the order of 0.964 to 1 between −40°C and 0°C was considered following Jouzel and Merlivat (1984), with a supersaturation parameter λ of 0.004 according to Risi (2010)."* Slightly changed figures and numbers throughout the manuscript in consequence of the change were updated.

9. p. 5, L. 13: I would recommend to say "examined transport conditions" or "examined moisture transport regions" or similar.

Our reply: We changed the sentence to *"[…] during the examined transport conditions"*.

10. p. 5, L. 22-25: I am not sure if I understand this paragraph correctly but it points to an important assumption and source of uncertainty of the moisture source diagnostics. Could the authors rephrase? Maybe just removing "In this case" and adding at the end of the last sentence ", only by the freshly evaporated moisture"?

Our reply: For clarification we changed the paragraph to: *"If the entrained moisture from evaporation was transported via small-scale turbulence to the trajectory altitude, mixing with air from below the trajectory level is likely. However, applying the above equations, changes of $R_D$ and q in consequence of mixing with low-level air masses are ignored. We therefore implicitly assume that the air masses below the trajectory level experienced a similar transport and precipitation history as the tracked air parcel. $R_D$ and q of the tracked air parcel are not affected by mixing with that air from below, only the by freshly evaporated moisture."*

11. p. 5, L. 30: Could the authors quantify the influence of non-equilibrium fractionation on δD? It should be around 1-3‰ depeding on the environmental conditions.

Our reply: We now consider a constant kinetic fractionation factor of 1.005 / 1.021 over the ocean / over land in the model. This changed the mean modeled δD by -0.7 /-4.7 ‰. Changed figures and numbers in consequence of the model changes were updated. For the analysis with respect to cold snaps the model changes are almost irrelevant as most air masses during the cold snaps originated from the continent and bare soil evaporation is ignored by the model in the case of $T_{skin}<0°C$.

12. p. 7, L. 1: No seasonal changes in FT?
Our reply: To the best of our knowledge the mean FT of Europe as well as the seasonality of FT are only roughly known. Estimates of the mean FT for Western Europe range from about 0.4 (Henderson-Sellers et al., 2006) to about 0.75 (Lawrence, Thornton, Oleson, & Bonan, 2007). Considering this large uncertainty of the mean FT, it is even more difficult to reliably estimate the seasonality of FT. We added a sentence to Section 4.2 stating the model sensitivity on the assumed FT: *"Increasing for instance the fraction of plant transpiration on total evapotranspiration from 0.7 to 0.8 in the model increases the mean modeled δD from summer by 2.0‰."*

13. In general: I would suggest not using italics for [18]O and D.
Our reply: changed

14. p. 8, L. 8: "ignored"? What does that mean? That it does not impact total uncertainty?
Our reply: We did not find the "ignore" the reviewer refers to, but we assume that the comment refers to kinetic fractionation. The model now considers kinetic fractionation during surface evaporation and under conditions of supersaturation in clouds.

15. p. 9, L. 8: "of the whole calibrations", maybe clearer if one said "found using all calibrations".
Our reply: We changed the sentence to *"Therefore, we only applied the average humidity dependence found using all calibrations […]"*

16. p. 9, L. 13: What does "WSW" mean?
Our reply: We changed the sentence to *"[…], which was mounted on a meteorological tower 30m above ground level 900m in the west-southwest."*

17. p. 10, L. 12: It would be nice to have the period that is covered by the measurements right at the beginning of Section 4.
Our reply: We changed the introduction of Section 4 to: *"In this section, we present the measurements from Karlsruhe, covering the time period from January 2012 to May 2013. For this time period, we identify specific circulation regimes related to cold snaps in Karlsruhe. Subsequent to this, we examine the capability of the Lagrangian isotope model of reproducing corresponding variations of δD."*

18. p. 11, L. 4: Fig 3c in JJA I do not see the continental effect so clearly, is it just the colorbar range or is it that in summer continental recycling smears out the strong continental gradient visible in winter. So should only a reference to Fig. 3b be given here?
Our reply: The continental effect is smaller in summer, likely because of smaller temperature gradients and the continental recycling. We removed the reference to Fig. 3c.

19. p. 12, L. 7: Shouldn't it say "Sublimation of snow or snow-melt evaporation?"?
Our reply: We changed the heading to: *"Sublimation of snow or snowmelt evaporation?"*

20. p. 12, L. 19: "To this end, we identified moisture uptake at $T_{skin}$< 0°C". This is a repetition the sentence before is enough in my opinion.
Our reply: We removed this sentence.

21. p. 14, L. 8: I assume the air masses originating from higher altitudes are also very dry so that the subsequent history of the air mass plays an important role and particularly subsequent uptakes so that I would not be so worried about the uncertainty associated with these air parcels' initialisation.
Our reply: We agree that model uncertainty corresponding to these trajectories is not a big problem because of the subsequent moisture uptake. To shorten discussion we, however, followed the suggestion of reviewer 2 and removed the 2% of air masses which during cold snaps originated from altitudes higher than 2km above ground level from the analysis.

22. p. 15, L. 18: In table 1 only ΔδD are stated and I would find it important to also mention the root mean square or absolute difference. Figure 9 of course also helps with respect to this point.
Our reply: We added the statistical uncertainty of the calculated mean differences to table 1 (root mean-square error divided by the square root of the number of observations).

**Table 1.** Average differences between modeled and measured $\delta$D of the 174 selected data points ($\Delta\delta$D), data points of group "cold" ($\Delta\delta D_{cold}$), and data points of group "warm" ($\Delta\delta D_{warm}$) from different model runs ($M$). $\pm$ states the statistical uncertainty of the averages (root mean-square error divided by the square root of the number of observations). Values of particular interest are printed in bold type.

| Name of model run | Description of model run | $\Delta\delta$D | $\Delta\delta D_{cold}$ | $\Delta\delta D_{warm}$ |
|---|---|---|---|---|
| $M_{MW}$ | fractionating evaporation of meltwater; (**reference run**) | **−18.6±1.5‰** | **−26.9±1.8‰** | **−10.4±2.1‰** |
| $M_{MW+,snow}$ | fractionating evaporation of meltwater; $\delta$D values of snow increased by 11.5‰ | −9.4±1.5‰ | −17.5±1.9‰ | −1.3±2.1‰ |
| $M_{MW+,ini}$ | fractionating evaporation of meltwater; $\delta$D values at initialization increased by 11.8‰ | −13.0±1.5‰ | −21.7±1.9‰ | −4.4±2.1‰ |
| $M_{MW+,upt.36h}$ | fractionating evaporation of meltwater; reduced moisture uptake in consequence of smoothing $q$ along the trajectories with a 36 h broad rectangle kernel (instead of 24 h) | −16.3±1.6‰ | −25.5±1.8‰ | −7.1±2.2‰ |
| $M_{MW+++}$ | fractionating evaporation of meltwater; simultaneous occurrence of the three assumptions above | −0.9±1.6‰ | **−10.5±1.8‰** | **+8.6±2.2‰** |
| $M_{S}$ | non-fractionating sublimation of snow | **+26.9±1.6‰** | +24.8±2.6‰ | +29.0±2.0‰ |
| $M_{S−,snow}$ | non-fractionating sublimation of snow; $\delta$D values of snow decreased by 11.5‰ | +17.1±1.6‰ | +14.8±2.5‰ | +19.4±2.0‰ |
| $M_{S−,ini}$ | non-fractionating sublimation of snow; $\delta$D values at initialization dencreased by 3.6‰ | +25.2±1.7‰ | +23.3±2.6‰ | +27.2±2.0‰ |
| $M_{S−,upt.12h}$ | non-fractionating sublimation of snow; increased moisture uptake in consequence of smoothing $q$ along the trajectories with a 12 h broad rectangle kernel (instead of 24 h) | +26.9±1.6‰ | +24.8±2.5‰ | +29.1±2.0‰ |
| $M_{S---}$ | non-fractionating sublimation of snow; simultaneous occurrence of the three assumptions above | **+16.0±1.6‰** | +13.9±2.4‰ | +18.2±2.0‰ |

23. p. 16, L. 8: °$C$ should be C
Our reply: We didn't find this typo.

24. p. 24, Figure 1: I find this Figure very helpful. I just did not understand whether the dashed blue and green lines indicate exact locations of precipitation/uptakes? If yes then why is the green line not continuous between -2.5 and -2 days? If no, then the authors should explain how the lines should be interpreted in the caption.
Our reply: Yes, the colors indicate exact locations of precipitation/moisture uptake. We replaced the dashed green line (long dashes) by a continuous line and clarified the meaning of the colors in the caption.

[Figure]

**Figure 3.** Illustration of the isotope modeling for one exemplary back trajectory (arrival in Karlsruhe March 18, 2012, 0 UTC). **(a)** Altitude of the back trajectory (black) and terrain height (gray/blue). The isotope model was initialized at $80°$ N in the marine boundary layer (MBL). In a low-pressure system near Iceland the tracked air parcel ascended to an altitude of 3200 m. During the last three days of transport to Karlsruhe the air parcel was sinking to the sampling altitude. **(b)** After initialization in the MBL, specific humidity $q$ (light blue and green colored line) and $\delta D$ values (thick black line) of the tracked air parcel were slightly decreasing, due to the formation of precipitation (dashed light blue lines) within the first day. More pronounced formation of precipitation, in consequence of lofting in a low-pressure system near Iceland, resulted in a second decrease of $q$ and the modeled $\delta D$ value dropped accordingly. Due to moisture uptake (green lines), related to a descent of the air parcel in the subsequent days, $q$ and the $\delta D$ value were increasing, until the air parcel reached Karlsruhe. Black thin curves illustrate the modeled $\delta D$ for different initializations of $\delta D$ (Sect. 2.3.3). The dependence on the initialization decreases with the amount of moisture uptake along the trajectories and is only low in Karlsruhe.

25. p. 27, Figure 5: state the period associated with these climatological source regions.
Our reply: We changed the first sentence in the caption to: *"Source regions of moisture (q) 30m above ground level in Karlsruhe (black star) based on five-day back trajectories for the time period January 2012 to May 2013."*

26. p. 29, Figure 8: Why is the total identified moisture along the trajectories only 48%? It seems low to me. Is it due to the trajectory length?

Our reply: The identified moisture fraction of 48% is a consequence of the chosen trajectory length of five days, and the smoothing of specific humidity along the back trajectories by 24h. If smoothing humidity along the back trajectories for 12 h instead of 24 h the identified humidity is 68%.

We clarified motivation of the smoothing in Section 2.1.: *"In addition, using GDAS wind fields with three-hourly time resolution for the calculation of back trajectories – which is a coarse time resolution compared to the GDAS time steps on the order of minutes – may cause deviations between the HYSPLIT back trajectories and the exact trajectories that air parcels followed in the GDAS. This, in turn, may result in artificial changes of q along the HYSPLIT back trajectories. For instance HYSPLIT trajectories not fully capturing diurnal vertical movement in consequence of thermal expansion of the atmospheric boundary layer in the GDAS may show artificial diurnal changes of q. Like Stohl and James (2004) and Sodemann et al. (2008b), we assume P and E to be the dominant processes and ignore the other effects. To avoid an overestimation of the formation of precipitation and moisture uptake in consequence of potentially artificial diurnal variations of q along the HYSPLIT trajectories, we smoothed q along the back trajectories with a 24 h rectangle function."*

As the smoothing may remove real diurnal variations of q in consequence of the formation of precipitation and moisture uptake, we assessed uncertainty regarding the smoothing by changing the width of the smoothing window and added the following sentences:

Section 4.1.: *"Because of the finite length of the five-day back trajectories, the total identified humidity is lower than 100%. If smoothing specific humidity along the back trajectories for 24 h, the total identified humidity accounts for 47%. If smoothing specific humidity for 12 h, diurnal and sub-diurnal variations in humidity are interpreted as the formation of precipitation and moisture uptake, which increases the total identified humidity to 63%."*

Section 5.1.: *"The mean fraction of moisture identified for these air masses by the moisture source diagnostic is 48%. If smoothing humidity along the back trajectories for 12 h instead of 24 h the mean identified fraction is 68%."*

Henderson-Sellers, A., Fischer, M., Aleinov, I., McGuffie, K., Riley, W. J., Schmidt, G. a., … Irannejad, P. (2006). Stable water isotope simulation by current land-surface schemes: Results of iPILPS Phase 1. *Global and Planetary Change*, *51*(1-2), 34–58. http://doi.org/10.1016/j.gloplacha.2006.01.003

[revised manuscript text omitted]

---

## Author Comment (AC2) · 4 Jan 2017

**Replies to reviewer 2**

**by E. Christner, M. Kohler, M. Schneider**

**January 4, 2017**

We like to thank reviewer 2 for the review and the constructive comments, which helped to improve the manuscript.

According to the comments of reviewer 1, we implemented (1) kinetic fractionation during surface evaporation at skin temperatures above 0°C and (2) reduced fractionation under conditions of supersaturation in clouds into the model. Because of this, most numbers and figures throughout the manuscript are slightly changed compared to the initial version. A marked-up manuscript version is attached at the end of this document.

**General comments:**

The manuscript by Christner et al. presents 17 months of new, continuous, low-level atmospheric water vapor δD measurements at a site near Karlsruhe, Germany in conjunction with a Lagrangian isotope model to inform understanding of the along-trajectory controls on measured δD variability. Integration of HYSPLIT-calculated, low-level back-trajectories with the isotope model shows that much of the observed variability in measured δD values is the result of identifiable along-transport processes most generally associated with moistening and dehydration of the air parcel through evapotranspiration and precipitation processes, respectively. A subset of backtrajectories associated with 'cold snaps' result from a systemic shift to continental source regions and easterly trajectories. For this subset, the authors investigate an additional controlling mechanism for isotopic evolution of these easterly trajectories, that of isotopic modification via moistening from surface snow sublimation. In the closing section of the manuscript, the authors investigate a range of possible conditions for isotopic modification of regional snowpacks that best explains measured δD, namely skin temperature controls on fractionating versus non-fractionating sublimation processes. From this, the authors determine the relevant skin temperature window for which post-depositional isotopic modification of snowpacks and associated impacts on low-level atmospheric moisture δD is most relevant.

This is a detailed and comprehensive manuscript that presents a new and robust long-term δD dataset that proves valuable for investigating controls on the isotopic evolution of low-level atmospheric moisture. The methods applied are appropriate. I particularly like the integration of HYSPLIT-derived trajectories into the new Lagrangian model presented and find the observed model δD congruence (Figs. 6 and 7) impressive and supporting of the Lagrangian-model approach. I find the discussion of the isotope effects of sublimation both nuanced and convincing, which is important given that this impactful process is generally neglected or assumed to be negligible in similarly-focused studies. I expect the findings and research design of this study will be of interest to a broad audience, particularly in light of the expansion of laser-based isotope analyzers that are sure to increase the number of similar isotope records in coming years. Accordingly, I strongly support publication of the manuscript in ACP pending some minor/moderate revisions to the text and some figures in order to (1) reduce redundancies in the text, (2) provide additional clarification for components of the HYSPLIT-Lagrangian isotope model integration and (3) ensure the more complex aspects of the manuscript are understandable to the broader audience likely to be interested in this work (e.g., earth scientists interested in proxy-based investigations of paleoaltimetry and paleoclimate).

**Specific comments:**

(1) *Reorganization and reducing redundancies* – much of the text in Section 3.3 is redundant with Section 2.3.2. It seems much of 3.3 could be moved and combined with 2.3.2. Similarly, section 3.4 falls under the heading 'Measurements'; however, this moisture source data seems more relevant to the model description in Section 2.3.

Our reply: We removed the redundancies by moving the information from sections 3.3 and 3.4 into the sections 2.3.1 and 2.3.2.

The opening paragraph of Section 2.3.3 presents some basic back-trajectory statistics but explanation is limited here. Important clarifying information is not provided until Section 4, specifically Section 4.3. I suggest saving back-trajectory statistics for Section 4 when more details needed for clarification are presented.
Our reply: We slightly shortened the trajectory statistics and moved it to Sect. 4.1.

Discussion of 24-hour smoothing window on page 4 could be removed there and saved for the same discussion on p. 14 (lines 19-28).
Our reply: We shifted information regarding model uncertainty from the smoothing to Sect. 5.1 (p.15). Since already the moisture source diagnostics in Sect. 4.1 and at the beginning of Sect. 5.1 refer to the smoothing of q, we added a clarifying motivation for the smoothing to Sect. 2.1.: *"In addition, using GDAS wind fields with three-hourly time resolution for the calculation of back trajectories – which is a coarse time resolution compared to the GDAS time steps on the order of minutes – may cause deviations between the HYSPLIT back trajectories and the exact trajectories that air parcels followed in the GDAS. This, in turn, may result in artificial changes of q along the HYSPLIT back trajectories. For instance HYSPLIT trajectories not fully capturing diurnal vertical movement in consequence of thermal expansion of the atmospheric boundary layer in the GDAS may show artificial diurnal changes of q. Like Stohl and James (2004) and Sodemann et al. (2008b), we assume P and E to be the dominant processes and ignore the other effects. To avoid an overestimation of the formation of precipitation and moisture uptake in consequence of potentially artificial diurnal and sub-diurnal variations of q along the HYSPLIT trajectories, we smoothed q along the back trajectories with a 24 h rectangle function."*

(2) *Model clarification and limitations* – Given the uncertainty in vapor δD at high altitudes (> 2 km) and low proportion of trajectories (2%) encountering these altitudes, it seems these trajectories could simply be removed from the analysis.
Our reply: We removed the 2% air masses during cold snap which originated from altitudes higher than 2km above ground level from the analysis and deleted the discussion of model uncertainty from model initialization at high altitudes from Sect. 5.1. We updated slightly changed numbers and figures resulting from this change and added two sentences to the second paragraph in Section 5.1: *"Only 2% of the corresponding air masses originated from altitudes higher than 2000m above ground level. Since uncertainty of model initialization is especially high for these air masses, we finally excluded the 2% of air masses originating from high altitudes."*

Beyond the vapor measurements at the Karlsruhe study site, are there other published datasets that would ground truth the isotope model presented? Specifically, are there any regional records of soil water δD (δ18O) that can be presented to constrain the RCWIP values used?
Our reply: We expanded the motivation for assuming the δD of soil water to be close to the GNIP based monthly RCWIP data (Sect. 2.3.2): *"Because soil water in Central Europe is generally frequently recharged by precipitation, we ignore systematic enrichment of $HD^{16}O$ in the uppermost soil layer caused by continuous fractionating evaporation from the soil. For instance measurements of precipitation amount at the measurement site in Karlsruhe indicate recharge of soil water around the site by more than 1mm precipitation per day on average every 2.9 days. The assumption regarding enrichment is supported by findings of (Risi et al., 2016), who observe insignificant systematic deviations of isotope ratios of water within the upper 15 cm of the soil from isotope ratios in precipitation at several sites in France, Germany, and the Czech Republic."*

Additionally, are there any regional snowpack δD (δ18O) records that would give a better understanding of the degree of snowpack δD variability? It is likely that snowpack δD varies both spatially and temporally throughout the accumulation and melt season in the study region, thus some discussion on how this variability limits the model presented is important.

Our reply: In general the GNIP based RCWIP data reflects the spatial variability of the δD of snowfall (Fig. 1b) and also accounts for changes of δD in different months of a year. We expanded the discussion regarding the δD of surface layer snow during the accumulation and melt season in Sect. 5.1 and added the following paragraph: *"In contrast to this, post-depositional fractionation processes may increase the δD values of surface layer snow on the order of +10 to +20‰ during periods with small accumulation rates (Gurney & Lawrence, 2004; Moser & Stichler, 1974), suggesting a scenario where the δD values of surface layer snow are higher than the δD from the RCWIP. Please note that this scenario is not likely for a seasonal snowpack during melt season since ablation may uncover old snow from colder winter months, causing changes of the δD of surface layer snow of −50‰ (Dahlke & Lyon, 2013). Given the relatively small snow depths in the investigated moisture source region (Fig. 8b), we ignore the uncovering of older snow layers and only consider a potential post-depositional increase of the δD of surface layer snow."*

Another limitation that might be more explicitly discussed or tied into the previous point about spatial δD variability of the snowpack is that of the 1°x1° resolution of the GDAS data set. How might this spatial smoothing impact ability to model δD variation?

Our reply: We added two paragraphs clarifying the meaning of $T_{subl,max}$ and the impact of horizontal resolution to Sect. 6: *"The determined $T_{subl,max}$ refers to a GDAS skin temperature that was weighted with positive latent heat flux at ground level. For this reason $T_{subl,max}$ is representative for GDAS skin temperatures during the day, when evaporation is strongest. However it should be kept in mind that due to the coarse resolution of 1×1° of the GDAS data, much spatial variability of the skin temperature is smoothed out. The meaning of the $T_{subl,max}$ derived in this study is therefore an average temperature in a 1×1° grid cell above which the evaporation of meltwater exceeds the amount of moisture from sublimation. A way to derive a more physical temperature separating the regimes of sublimation and meltwater evaporation could be using data with a higher horizontal resolution. This wouldn't necessarily improve accuracy with respect to the back trajectories' positions but locations with enhanced skin temperatures and especially high amounts of surface evaporation would be more realistically represented, presumably resulting in higher evaporation weighted skin temperatures and a higher $T_{subl,max}$.*
*In addition, higher horizontal resolution would allow to better account for spatial heterogeneity of the δD of surface layer snow in mountainous regions, for instance by also weighting the δD of the snow with the amount of surface evaporation. In this context please note that systematic uncertainty regarding the δD of surface layer snow turned out to be the main limitation for determining $T_{subl,max}$ with the approach presented here. For this reason regular analysis of the δD of surface layer snow samples for instance at selected GNIP stations would be a very desirable and efficient measure to reduce uncertainty of $T_{subl,max}$."*

(3) *Clarifying complexity* –I find Section 5.2 and corresponding Figure 10 difficult to comprehend. I understand the general idea that multiple model runs were used to identify the cutoff temperature between fractionation and non-fractionating sublimation, but is not clear to me how the 16 scenarios of 'side constraint' variability and associated 128 total model runs correspond to the lines shown in Fig. 10. How do 128 total model runs translate to 9 distinct lines in Fig. 10? Please clarify in the text and figure caption and reconsider what Figure 10 should show to more clearly communicate the information in this section.

Our reply: Figure 10 shows the results from 144 model scenarios with 16 different $T_{subl,max}$ and the 9 configurations $M_{S\_MW}$, $M_{S\_MW,-,snow}$, $M_{S\_MW,-,ini}$, $M_{S\_MW,-,upt.12h}$, $M_{S\_MW,---}$ , $M_{S\_MW,+,snow}$, $M_{S\_MW,+,ini}$, $M_{S\_MW,+,upt.36h}$, $M_{S\_MW,+++}$. For clarification we adapted Figure 10 and the respective caption and added some further explanation to Sect. 5.2 : *"To assess this uncertainty of the optimal $T_{subl,max}$, we performed 16·8=128 additional model runs (Fig. 10, thin lines) corresponding to 16 different $T_{subl,max}$ from −15 to 0°C and 8 different model configurations with the same assumptions about a changed δD of snow, a changed δD at the model initialization, a different amount of moisture uptake, and superposition of the three effects as in the above uncertainty assessment for the $M_S$ and the $M_{MW}$. Analogous to the uncertainty assessment for the $M_S$ and the $M_{MW}$, we refer to the model runs as $M_{S\_MW,+,snow,T_{subl,max}}$, $M_{S\_MW,+,ini,T_{subl,max}}$ , $M_{S\_MW,+,upt.36h,T_{subl,max}}$ , $M_{S\_MW,+++,T_{subl,max}}$, $M_{S\_MW,snow,T_{subl,max}}$, $M_{S\_MW,-,ini,T_{subl,max}}$, $M_{S\_MW,-,upt.12h,T_{subl,max}}$ , and $M_{S\_MW,---,T_{subl,max}}$."*

[Figure]

**Figure 10.** Mean difference between modeled and observed δD of water vapor at Karlsruhe for selected cold snap events (ΔδD). Each cross represents the ΔδD obtained from one model configuration with specific model assumptions regarding the $T_{subl,max}$, the δD of snow, the δD at the model initialization, and the amount of moisture uptake. The thick black line connects the ΔδD from 16 model configurations with $T_{subl,max}$ between −15 and 0°C and the model assumptions of $M_{S\_MW,T_{subl,max}}$ and illustrates the increase of the modeled δD values with increasing maximum skin temperature allowing non-fractionating sublimation. The gray shaded area depicts possible systematic changes of the ΔδD in the case of a systematic deviation of the δD of snow from the RCWIP climatology (light blue), a systematically changed δD at the model initialization (yellow), a systematically changed amount of moisture uptake (green), and superposition of the different assumptions (thin black lines). The thin dashed black line corresponds to the model configuration $M_{S\_MW,+++,T_{subl,max}}$, which doesn't allow reproducing the observations in group "warm" and is therefore not considered in the gray shaded area of uncertainty. Magenta: optimal $T_{subl,max}$ for best agreement of model and observation (dot) and uncertainty of the optimal $T_{subl,max}$ (error bar) due to uncertainty of model assumptions.

Given the focus on easterly trajectories, a figure more clearly showing association of easterly trajectory pathways with corresponding snow cover in that region would be helpful. This might be accomplished by adding a panel to Figure 8 showing snow cover.

Our reply: We added a second panel to Figure 8 showing the mean GDAS snow height during the cold snaps.

[Figure]

**Figure 8. (a)** Lagrangian source region analysis of low-level water vapor in Karlsruhe for the observed cold snaps. The mean identified fraction of moisture along the five-day back trajectories is 48%. **(b)** Mean snow depth during the cold snaps based on GDAS data.

(4) The title, introduction, and conclusions sections place focus exclusively on the sublimation aspects of this work. I think this undersells and undervalues the importance of the broader Lagrangian isotope model approach and its applications. Sublimation appears to only relate to the easterly trajectory subset (< 50% of trajectories investigated). I am not sure if there is a companion paper planned/submitted/published for the (north)westerly trajectories but these trajectories seem important to discuss in more detail as well, even with a single summary paragraph somewhere in Section 4 of what was learned from modeling these trajectories. If manuscript focus is to be exclusively on sublimation effects, as the title implies, I don't think all trajectory data should be included (i.e., Fig. 5) and the δD record should focus on 'cold snap' trajectories. Currently, there seems to be too much data shown in Figures 5-7 that receive too cursory of a discussion.

Our reply: We added the following paragraph to Sect. 4.2: "*With respect to surface evaporation the air masses from the West and from the East contain very different types of information. The air masses from the West are exposed to moisture uptake from the ocean and to continental evapotranspiration at relatively warm temperatures. Therefore they contain information about isotope fractionation during evaporation from the ocean, evaporation from warm land surfaces, and plant transpiration. This makes the modeled δD of westerly air masses especially sensitive to simplifying model assumptions regarding the δD of moisture from continental evapotranspiration at warm skin temperatures. Increasing for instance the fraction of plant transpiration on total evapotranspiration from 0.7 to 0.8 in the model increases the mean modeled δD from summer by 2.0‰. Assuming δD values of soil water which are systematically increased by +10‰ increases the mean modeled δD from summer by +2.4‰. In contrast to the westerly air masses, the easterly air masses during cold snaps are sensitive to isotope fractionation during surface evaporation at temperatures where snow and meltwater exist. For these easterly air masses assumptions regarding evapotranspiration at warm skin temperatures only have a small impact.*"

We replaced the seasonal panels in Fig. 5 by one source region map for the whole time period covered by the isotope observations in Karlsruhe and shortened the respective caption. In addition, we shortened Section 4.1 and combined it with Section 4.2. In Fig. 6 we kept the whole data for demonstrating the exceptional temperature and δD during the cold snaps. In Fig. 7 we also kept all data, because it underlines the capability of the model of reproducing observations over a wide range of temperatures and for different synoptic conditions.

[Figure]

**Figure 5.** Source regions of moisture ($q$) 30 m above ground level in Karlsruhe (black star) based on five-day back trajectories for the time period January 2012 to May 2013. The color code indicates the contribution of different source regions to $q$ in Karlsruhe in % per km$^2$. Integration over the whole map gives the total identified humidity of 47%.

**Technical comments**

*p. 1, Line 16:* 'isotopologues' of water more appropriate since molecules are listed
Our reply: changed

*p. 2, Lin4 4:* somewhat misleading to say snowpacks are most sensitive to isotopic modification. Lakes, for example, generally provide longer-duration integration of post-depositional modification.
Our reply: We changed this sentence to "*Most problematic in that context are systematic post-depositional changes of isotope ratios into one direction, which most likely affect water reservoirs with long exposure to the atmosphere such as water intercepted in the snowpack.*"

*p. 2, Lines 7-8:* Here and throughout I think 'fractioning' should be replaced with 'fractionating'. I am unfamiliar with the term 'fractioning' with regards to isotopes.
Our reply: changed

*p. 2, Line 8:* clarify what is meant by 'other part'
Our reply: We changed the sentence to "*To explain the observations, the authors suggested non-fractionating sublimation with part of the sublimated vapor fractionating recondensing and part of the sublimated vapor escaping the snow layer.*"

*p. 2, Line 13:* It might be helpful to introduce the delta notation earlier in the manuscript so that it can be used here, i.e., 'increase of δ values'
Our reply: We moved the definition of δD into the introduction: "*In consequence, not only specific humidity and dew point temperature but also the isotope concentration ratio $R_D=[HD^{16}O]/[H_2^{16}O]$ – commonly referred to as $\delta D=R_D/R_{D,VSMOW}-1$ with $R_{D,VSMOW}=0.00031152$ – decreases in a cooling and raining air mass.*"

*p. 2, Line 21:* What is meant by 'resublimation'?
Our reply: Corrected. What we meant is *"deposition"*.

*p. 4, Lines 12-13:* Quantify what is meant by 'fast' – sub-diurnal?
Our reply: We replaced 'Fast and random variations' (now on p.15) *by "Potentially artificial diurnal variations […]"* and 'may smooth out real diurnal variations' (now on p.15) *by "[…] may smooth out real sub-diurnal and diurnal variations […]"*

*p. 4, Line 15:* What is meant by 'displacement' of moisture uptake amount?
Our reply: We mean by 'displacement' that pronounced moisture uptake at a certain location is interpreted as continuous moisture uptake over a longer time interval in consequence of smoothing q. As we added more expanded clarifying information regarding the smoothing, this seems to be clear enough. So we removed 'displacement' from the sentence (now on p.15): "*Arbitrarily choosing a width of 24 h may smooth out real sub-diurnal and diurnal variations of q and thereby, may result in an underestimation of the amount of moisture uptake.*"

*p. 5, Line 3:* I think the phrasing 'more likely' should be changed – 'preferential fractionation of D into the liquid phase' is more precise language.
Our reply: We changed the respective sentence to "*Because of preferential fractionation of D into the liquid phase, the formation of precipitation results in a decreasing isotope concentration ratio ($R_D=[HD^{16}O]/[H_2^{16}O]$) in an air mass.*"

*p. 5, Line 25:* Clarify what is meant by 'not affected by dilution'
Our reply: For clarification we changed the paragraph to "*If the entrained moisture from evaporation was transported via small-scale turbulence to the trajectory altitude, mixing with air from below the trajectory level is likely. However, applying the above equations, changes of $R_D$ and q in consequence of mixing with low-level air masses are ignored. We therefore implicitly assume that the air masses below the trajectory level experienced a similar transport and precipitation history as the tracked air parcel. $R_D$ and q of the tracked air parcel are not affected by mixing with that air from below, only the by freshly evaporated moisture.*"

*p. 6, Line 15:* Where does the 2.6 day value come from?
Our reply: Ten-minutely measurements of precipitation amount at the measurement site indicate precipitation with more than 0.1mm per ten minutes on average every 2.6 days. As precipitation per day might be a more intuitive unit, we changed the sentence to: "*For instance measurements of precipitation amount at the measurement site in Karlsruhe indicate recharge of soil water around the site by more than 1mm precipitation per day on average every 2.9 days.*"

*p. 11, Line 1:* Here and elsewhere, should use 'value' instead of 'ratio' to describe δD
Our reply: replaced

*p. 11, Line 4:* I think the statement 'caused by the relation between δD and condensation temperature' is misleading. The more direct control on the continental effect is degree of rainout.
Our reply: We replaced 'condensation temperature' by *"degree of rainout"*

*p. 12, Lines 18-19:* Redundant sentences: 'Second….' And 'To this end….'
Our reply: We removed the two sentences.

*p. 12, Lines 20-21:* Rationale for only considering trajectories above 28% median level needs to be more clearly communicated.
Our reply: We clarified rationale for only considering these trajectories: "*The air masses with the smallest moisture uptake within the last five days are least sensitive to the δD of moisture from surface evaporation and therefore don't allow to robustly evaluate the model description of isotope fractionation during the sublimation of snow sublimation and the evaporation of meltwater. For a meaningful interpretation, we therefore only used the half of data with a contribution from surface evaporation at $T_{skin}<0$ higher than 28%.*"

*p. 13, Lines 4-5:* The term 'side constraints' here and following is unclear to me. Can you clarify what is meant by 'side'?
Our reply: We replaced the term "side constraints" by *"model assumptions"*.

*p. 15, Line 29:* Clarify what 'both groups' refer to.

[revised manuscript text omitted]